# Structural and dynamic insights into the activation of the μ-opioid receptor by an allosteric modulator

Shun Kaneko [1,2], Shunsuke Imai [1] ✉, Tomomi Uchikubo-Kamo [1], Tamao Hisano[1], Nobuaki Asao[1,2], Mikako Shirouzu [1] & Ichio Shimada [1,3] ✉

G-protein-coupled receptors (GPCRs) play pivotal roles in various physiological processes. These receptors are activated to different extents by diverse orthosteric ligands and allosteric modulators. However, the mechanisms underlying these variations in signaling activity by allosteric modulators remain largely elusive. Here, we determine the three-dimensional structure of the μ-opioid receptor (MOR), a class A GPCR, in complex with the $G_i$ protein and an allosteric modulator, BMS-986122, using cryogenic electron microscopy. Our results reveal that BMS-986122 binding induces changes in the map densities corresponding to R167[3.50] and Y254[5.58], key residues in the structural motifs conserved among class A GPCRs. Nuclear magnetic resonance analyses of MOR in the absence of the $G_i$ protein reveal that BMS-986122 binding enhances the formation of the interaction between R167[3.50] and Y254[5.58], thus stabilizing the fully-activated conformation, where the intracellular half of TM6 is outward-shifted to allow for interaction with the $G_i$ protein. These findings illuminate that allosteric modulators like BMS-986122 can potentiate receptor activation through alterations in the conformational dynamics in the core region of GPCRs. Together, our results demonstrate the regulatory mechanisms of GPCRs, providing insights into the rational development of therapeutics targeting GPCRs.

G-protein-coupled receptors (GPCRs) are seven-transmembrane receptors that play critical roles in mediating cellular responses. They function by transducing extracellular signals into cellular responses through interaction with various signaling partners on the intracellular side, such as heterotrimeric G proteins and arrestins. Given that over 30% of currently used therapeutics target GPCRs, these receptors are not only of biological importance but also of substantial pharmaceutical relevance[1].

Previous structural studies of GPCRs in various ligand-bound states have identified structural motifs such as DRY, PIF, and NPxxY, conserved across GPCRs[2,3]. Comparisons of these structures have provided detailed snapshots within the activation process and suggested structural changes in these motifs as well as an outward shift of the intracellular half of the 6th transmembrane helix (TM6), which opens the intracellular cavity to interact with intracellular effectors, during activation. Furthermore, previous functional studies on regulatory mechanisms of GPCRs demonstrated that the activity induced by orthosteric ligands—compounds that bind to the ligand-binding site of a receptor—does not necessarily represent the full extent of GPCR function. For example, the introduction of point mutations, not at residues that directly interact with the intracellular effectors, can increase the activities of GPCRs, suggesting that compounds that

[1]Center for Biosystems Dynamics Research (BDR), RIKEN, Kanagawa, Japan. [2]Graduate School of Pharmaceutical Sciences, The University of Tokyo, Tokyo, Japan. [3]Graduate School of Integrated Science for Life, Hiroshima University, Hiroshima, Japan. ✉e-mail: shunsuke.imai.ku@riken.jp; ichio.shimada@riken.jp

induce changes in GPCRs similar to these mutations could potentially increase their activities[4–6]. However, despite intensive screening, the discoveries of orthosteric ligands with greater efficacies than those exhibiting the highest efficacy levels have been rare[7,8], suggesting a limit to the level of GPCR activity that can be activated by orthosteric ligands.

Interestingly, allosteric modulators, compounds that bind to receptor sites different from the orthosteric ligand-binding site, have been shown to enhance GPCR activation beyond what is achievable by orthosteric agonists alone[9]. Although there are examples that have studied the mechanism of allosteric modulation of GPCRs using in silico approaches[10–13], the structural basis underlying the activity enhancement by the allosteric modulators remains incompletely understood. This gap in knowledge could potentially hinder the rational design of allosteric modulators. One reason for this limited understanding is the minimal structural differences observed when comparing GPCR three-dimensional structures in the presence and absence of allosteric modulators, although they provide valuable information on the interaction sites of the allosteric modulators on GPCRs[14]. The lack of discernible differences that can be readily connected to the function of GPCR might be because these structural characterizations were conducted in crystals or ice, conditions that suppress functional dynamics essential for GPCR functions, and/or because the binding of intracellular binders forces the structures of GPCRs into the intracellular effectors-bound forms, obscuring the possible differences that may exist in the absence of these intracellular binders, which directly dictate the differences in activities[1].

Within the class A GPCR family, the μ-opioid receptor (MOR) has garnered considerable interest due to its central role in analgesia and its susceptibility to opioid analgesics such as morphine[15]. Potential modulation of MOR function presents an avenue towards distinct analgesics that could avoid the issues associated with current opioid treatments such as dependency, tolerance, and constipation. BMS-986122 stands out as a subtype-selective allosteric modulator of MOR, capable of enhancing endogenous opioid peptide pathways[16–18]. In preceding studies[4,19], we used nuclear magnetic resonance (NMR) spectroscopy to uncover a three-state conformational equilibrium at the intracellular half of the transmembrane helix 6 (TM6) of MOR, in the absence of intracellular binders. This conformational equilibrium controls the activity of MOR, as shown by the correlation of the populations of the three interchanging conformations with the activity. From this correlation, we designated these conformations, possessing high, intermediate, and no relative activities, as the fully-activated, partially-activated, and inactivated conformations, respectively[19]. In the absence of G proteins, these conformations are under equilibrium, where the populations are changed depending on the efficacies of the ligand bound to the extracellular ligand-binding pocket. Furthermore, we also demonstrated that the intracellular half of TM6 is shifted outward in the fully-activated conformation in the absence of G proteins as in the GPCR structures bound to their cognate G proteins[19]. The observation of multiple states in the absence of G proteins, where some of them structurally resemble the G protein-bound states, has been observed for some other GPCRs[20–25], indicating a common mechanism of activation shared in class A GPCRs. It was also shown that BMS-986122 increases the population of the fully-activated conformation in the three-state conformational equilibrium, thereby enhancing MOR activity. The hydrophobic interactions between TM3 and TM5 are involved in the destabilization of the partially-activated conformation, a conformation where TM6 is shifted inward. However, since the hydrophobic interactions are among residues that are not conserved in GPCRs, the generalization of this notion to other GPCRs has been difficult. In addition, the interpretation of hydrophobic interactions is not straightforward[26,27], making it difficult to rationally design compounds that modulate the hydrophobic interface to enhance the activity of GPCRs.

Here, we determine the three-dimensional structure of MOR in complex with the heterotrimeric $G_i$ protein, in a full agonist [d-Ala2,N-MePhe4,Gly-ol5]enkephalin (DAMGO)- and the allosteric modulator BMS-986122-bound state, using cryo-electron microscopy (cryo-EM). Comparison of the two density maps in the presence and absence of BMS-986122 identifies a small difference induced by BMS-986122 binding, between the side chains of highly conserved residues R167[3.50] in the DRY motif and Y254[5.58] involved in the water-mediated interaction with the NPxxY motif (superscripts following residue numbers indicate the Ballesteros-Weinstein numbering for GPCRs[28]). We then use solution NMR spectroscopy to investigate the interaction between residues R167[3.50] and Y254[5.58], in the absence of the G protein, where MOR is fully functional. From the linewidth analyses of the methyl group of M257[5.61], it is demonstrated that R167[3.50] and Y254[5.58] interact transiently in the absence of BMS-986122, and this interaction is more populated in the presence of BMS-986122. Furthermore, it is shown that the interaction between R167[3.50] and Y254[5.58] works as a lock to stabilize TM6 at the outward-shifted position, increasing the population of the fully-activated conformation. Since R167[3.50] and Y254[5.58] are highly conserved among class A GPCRs, uncovering the importance of the interaction between these residues in their functions may substantially influence the rational design and development of allosteric modulators of MOR and other GPCRs.

## Results

### Cryo-EM structure of MOR(DAMGO)-$G_i$-scFv16 in complex with BMS-986122

In this study, we utilized an N-terminally BRIL-tagged[29,30] human MOR construct with an F158[3.41]W mutation[31,32], the heterotrimeric $G_i$ protein, and a single-chain antibody scFv16[33] for the cryo-EM study, and a human MOR construct with F158[3.41]W and six methionine substitutions, referred to as MOR/Δ6M, for the NMR study[4,19] (see "Methods" for details). Hereinafter, we designate the orthosteric ligand bound to MOR samples in parentheses, e.g., MOR in the full agonist DAMGO-bound state as MOR(DAMGO).

First, using cryo-EM, we analyzed the structure of the MOR(DAMGO)-$G_i$-scFv16 complex in the BMS-986122 bound state, at 3.1 Å resolution (Fig. 1, Supplementary Fig. 1, and Supplementary Table 1). For a direct comparison with the structure in the presence of BMS-986122, we also obtained a cryo-EM structure of the human MOR(DAMGO)-Gi-scFv16 complex in the absence of BMS-986122 at 3.0 Å resolution (Supplementary Fig. 2 and Supplementary Table 1), which was virtually identical to the previously reported structure of mouse MOR without the N-terminal BRIL tag[33] with the root mean square deviation (r.m.s.d.) of the Cα atoms of 0.71 Å (Supplementary Fig. 3). Then, by comparing the two maps of the MOR(DAMGO)-$G_i$-scFv16 complex in the presence and absence of BMS-986122, it was demonstrated that these two density maps are almost identical, as has been observed for other cases where cryo-EM structures of other GPCRs in complex with their allosteric modulators are reported[14]. Additionally, we observed density on the membrane-facing surface on TM3, 4, and 5 of MOR, only for the map of the MOR(DAMGO)-$G_i$-scFv16 complex obtained in the BMS-986122 bound state (Fig. 1), indicating that this density corresponds to BMS-986122 bound to MOR. The position of this additional density is consistent with our previous result that the substitution of T162[3.45] on TM3 to a methionine abrogated the binding of BMS-986122[19], supporting the notion that the aforementioned additional density originates from BMS-986122 bound to MOR (Fig. 1b). However, due to the relatively weak EM density at this allosteric site, as well as the moderate resolution of 3.1 Å, it was not possible to unambiguously model BMS-986122 solely by the cryo-EM data. To determine the orientation of BMS-986122 relative to MOR, we compared the NMR spectra of MOR(DAMGO) bound to

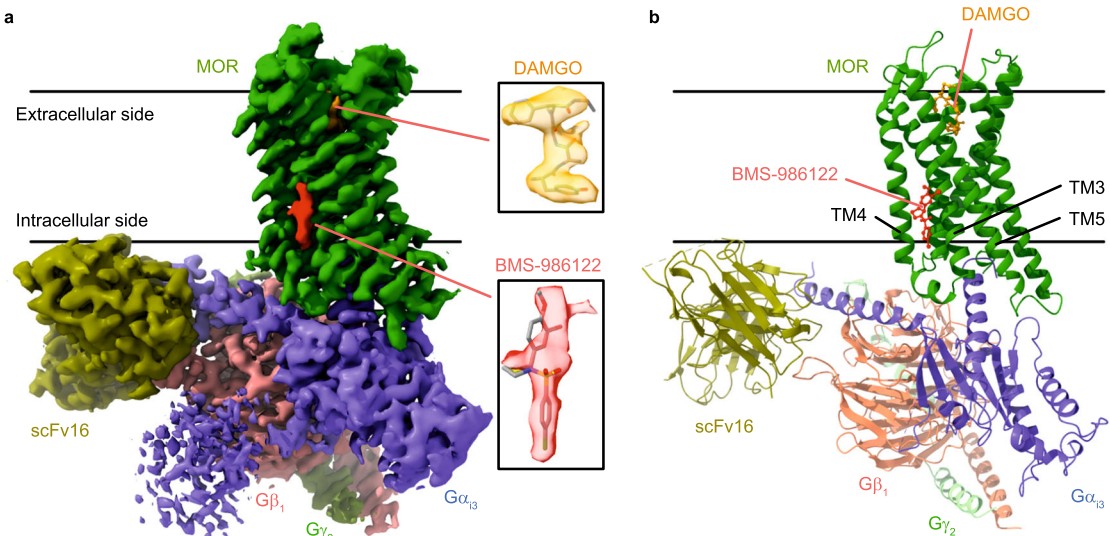

**Fig. 1 | Overall structure of MOR in complex with a full agonist and an allosteric modulator. a** A cryo-EM density map of the MOR(DAMGO)-$G_i$-scFv16 complex in the presence of BMS-986122. Insets show expanded views of the DAMGO and BMS-986122 models and their corresponding density maps. Densities are contoured at a threshold level of 2.4 in ChimeraX version 1.7. **b** The overall structure of the MOR(DAMGO)-$G_i$-scFv16 complex in the presence of BMS-986122.

BMS-986122 and its analog BMS-986124 (Supplementary Fig. 4). BMS-986124 is an analog of BMS-986122, in which the positions of the bromo- and methoxy-moieties on the benzyl group are different, that competes with BMS-986122 on MOR but without allosteric modulation activity (Supplementary Fig. 4b)[16]. When the $^1$H-$^{13}$C heteronuclear multiple quantum coherence (HMQC) NMR spectra of MOR(DAMGO) in the presence of BMS-986122 and BMS-986124 were compared, the NMR signal from M245$^{5.49}$ was largely perturbed; it was broadened beyond detection upon the addition of BMS-986122[19], whereas it was observed with a chemical shift perturbation upon the addition of BMS-986124 (Supplementary Fig. 4c). These results indicate that the methoxybenzyl bromide moiety of BMS-986122 is adjacent to M245$^{5.49}$, as modification of this moiety most largely perturbed the signal of M245$^{5.49}$.

We then docked BMS-986122 in the cryo-EM structure so that the methoxybenzyl bromide moiety is adjacent to M245$^{5.49}$ (Supplementary Fig. 4d). In the structure, the aromatic residues W158$^{3.41}$ on TM3, and F180$^{ICL2}$ in the intracellular loop 2 (ICL2) contact the aromatic rings of BMS-986122 (Supplementary Fig. 5a). To test whether these aromatic residues are involved in the interaction with BMS-986122, we analyzed the perturbation of the NMR signal of M245$^{5.49}$ for W158$^{3.41}$M or F180$^{ICL2}$L variants upon the addition of BMS-986122, and the allosteric modulation effects of these variants (Supplementary Fig. 5b, c). As mentioned above, the M245$^A$ signal is broadened upon the addition of BMS-986122 in the parental MOR/Δ6M(DAMGO), while in the F180$^{ICL2}$L or W158$^{3.41}$M variants, the M245$^A$ signal did not disappear in the presence of 100 μM BMS-986122, indicating that these variants possess reduced affinities for BMS-986122 compared to the parental MOR/Δ6M (Supplementary Fig. 5b). These results indicate that the side chains of F180 and W158 are involved in the interaction with BMS-986122, which supports the cryo-EM structure of the MOR(DAMGO)-$G_i$-scFv16 complex in the BMS-986122 bound state. This is further confirmed by a GTP turnover assay that measures the consumption of GTP by G proteins through GTP to GDP turnover[34–36] (Supplementary Fig. 5c). F180$^{ICL2}$L exhibited a loss of BMS-986122-dependent increase in the activity of MOR, while W158$^{3.41}$M exhibited an increase in the activity upon the addition of BMS-986122 as the parental MOR/Δ6M, suggesting that F180$^{ICL2}$ is more important for the allosteric modulation activity of BMS-986122 than aromatic residues at 3.41.

By comparing structural models of MOR(DAMGO) with and without BMS-986122, it has been shown that the positions of the main chain atoms are nearly identical (Fig. 2a, r.m.s.d. = 0.36 Å for all Cα atoms of MOR). However, notable differences between the density maps in the absence and presence of BMS-986122 were observed between the side chains of MOR residues in the structural motifs that have been shown to play important roles in the activation of GPCRs (Fig. 2b). These are R167$^{3.50}$ in the DRY motif, which is involved in the interaction with the $G_i$ protein, and Y254$^{5.58}$, a conserved residue involved in water-mediated polar contact with Y338$^{7.53}$ in the NPxxY motif (Supplementary Fig. 6). The side chains of these residues were modeled closer to each other in the cryo-EM structure of MOR(DAMGO)-$G_i$-scFv16 in the BMS-986122 bound state than in the BMS-986122 unbound state, as evidenced by a connecting density between these residues, which was observed in the cryo-EM map of the BMS-986122 bound state but not in the unbound state (Fig. 2b, red circles). These results suggested that the interaction between R167$^{3.50}$ and Y254$^{5.58}$ may underlie the activation of MOR by BMS-986122.

## Dynamic interaction between R167$^{3.50}$ and Y254$^{5.58}$ side chains in MOR

To investigate whether the structural differences between Y254$^{5.58}$ and R167$^{3.50}$ observed in the cryo-EM maps of MOR(DAMGO) with and without BMS-986122 in the $G_i$ protein-bound state (Fig. 2b) are relevant to the function of MOR, we conducted NMR analyses of MOR/Δ6M(DAMGO) in the absence of the $G_i$ protein, where it undergoes the conformational equilibrium that regulates the activity of MOR[19]. To this end, we observed the methyl NMR signal from M257$^{5.61}$, which resides next to Y254$^{5.58}$ on the same side of the TM5 helix (Fig. 3a). It has been shown that the linewidths of the NMR signals include the effect of chemical exchange of the probe among conformational substates, and the contribution of the exchange can be evaluated by the magnetic field dependence of the linewidths[37–40]. The $^{13}$C linewidths of M257$^{5.61}$ in MOR/Δ6M(DAMGO) in the absence of BMS-986122 were 65 ± 5 and 123 ± 8 Hz at 18.8 T (800 MHz $^1$H frequency) and 23.5 T (1 GHz $^1$H frequency), respectively, whereas those in the BMS-986122 bound state were 89 ± 7 and 109 ± 12 Hz at 18.8 and 23.5 T, respectively (Fig. 3b). The differences in the $^{13}$C linewidths at the two magnetic fields, 58 ± 13 and 20 ± 19 Hz in the absence and presence of BMS-986122, respectively, indicate that M257$^{5.61}$ undergoes a chemical exchange process among a set of different conformational substates in the μs-ms

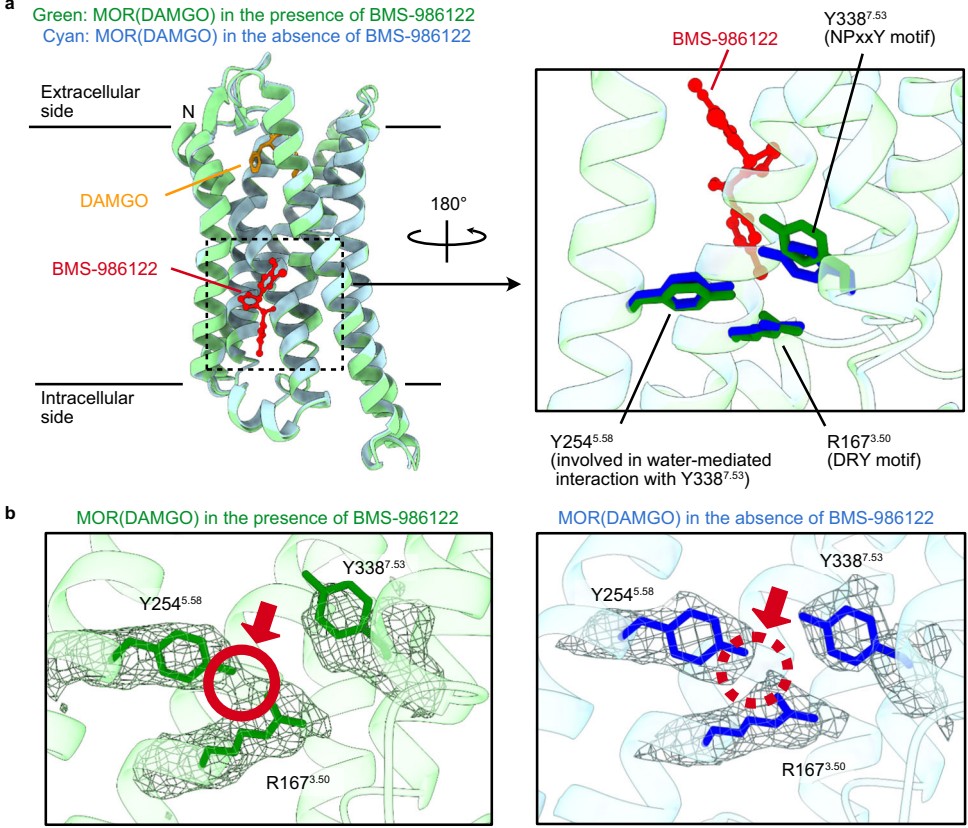

**Fig. 2 | Comparison of structures of MOR(DAMGO) in the presence and absence of BMS-986122. a** Comparison of cryo-EM structures of the MOR(DAMGO)-G$_i$-scFv16 complex in the presence (green) and absence (cyan) of BMS-986122. **b** Overlay of cryo-EM density maps and structures around R167[3.50], Y254[5.58], and Y338[7.53] residues of MOR(DAMGO) in the presence (left) and absence (right) of BMS-986122. The differences in density between R167[3.50] and Y254[5.58] are highlighted with red circles. Densities are contoured at 3.00 and 0.012 in ChimeraX version 1.7 for the maps in the presence and absence of BMS-986122, respectively, representing a significant difference in the contact region between R167[3.50] and Y254[5.58].

timescale, and that the transition is suppressed upon the addition of BMS-986122 (Fig. 3c).

We then tested whether this chemical exchange process among conformational substates observed for M257[5.61] is related to the interaction of the neighboring R167[3.50] and Y254[5.58], whose densities were different in the cryo-EM structures in the absence and presence of BMS-986122. To this end, we substituted Y254[5.58] with phenylalanine, which lacks the hydroxy group critical for the possible hydrogen bond interaction with R167[3.50]. Interestingly, M257[5.61] in the MOR/Δ6M/Y254[5.58]F(DAMGO) variant did not exhibit any signal in the $^{1}$H-$^{13}$C HMQC spectra both in the absence and presence of BMS-986122, due to line broadening (Fig. 3d). These results indicate that the transient hydrogen bond interaction of the hydroxy group of Y254[5.58] underlies the chemical exchange process observed for M257[5.61], and the binding of BMS-986122 enhances the interaction, which results in suppressing the chemical exchange process.

## Functional and structural role of the interaction between R167[3.50] and Y254[5.58]

In the crystal structure of antagonist-bound MOR[41], R167[3.50] and Y254[5.58] residues are separated, with the intracellular half of TM6 residing between these two residues due to the inward shift of the helix (Fig. 4a). In contrast, in the cryo-EM structure of the MOR(DAMGO)-G$_i$-scFv16 complex in the BMS-986122 bound state, the side chain atoms of R167[3.50] and Y254[5.58] fill the space formed by the outward shift of the intracellular half of TM6. We previously reported that, in the absence of G$_i$ protein, the intracellular half of TM6 undergoes a function-related conformational equilibrium among the inactivated, partially-

activated, and fully-activated conformations, where TM6 is outward-shifted only in the fully-activated conformation[19]. Taken together, these observations suggest that the hydrogen bond interaction between Y254[5.58] and R167[3.50] might stabilize the outward-shifted TM6 in the absence of the G protein, increasing the population of the fully-activated conformation.

We then acquired the $^{1}$H-$^{13}$C HMQC spectra of MOR/Δ6M(DAMGO) and analyzed NMR signals from the methyl group of M283[6.36], which is located at the intracellular tip of TM6 (Fig. 4a) and exhibits different chemical shift values in the inactivated, partially-activated, and fully-activated conformations, serving as an ideal probe for the quantification of the populations of the three conformations (Fig. 4b and Supplementary Fig. 7). In the absence of BMS-986122, two NMR signals from M283[6.36], reflecting the partially-activated and inactivated conformations were observed, while in the presence of BMS-986122, two NMR signals from M283[6.36], which reflect the fully-activated and inactivated conformations were observed (Fig. 4b). For the Y254[5.58]F variant, in which the interaction between R167[3.50] and Y254[5.58] would be abolished, the signals from the fully-activated and inactivated conformations decreased and increased, respectively, both in the absence and presence of BMS-986122. For the R167[3.50]L variant, in which the interaction between R167[3.50] and Y254[5.58] would also be abolished, similar effects were observed; i.e., the signals from both the fully-activated and partially-activated conformations decreased, whereas the signal from the inactivated conformation increased (Supplementary Fig. 8a). Together, these findings indicate that the disruptions of the interaction between R167[3.50] and Y254[5.58] by introducing the Y254[5.58]F or R167[3.50]L

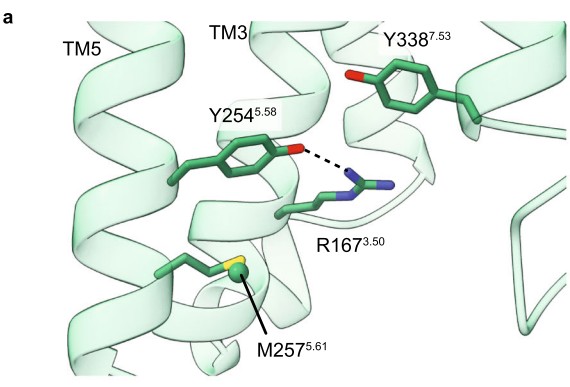

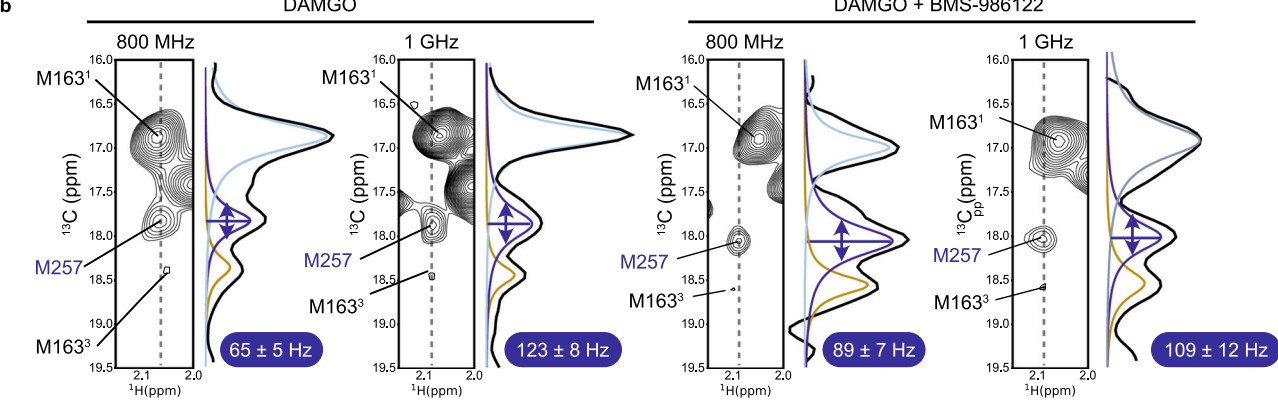

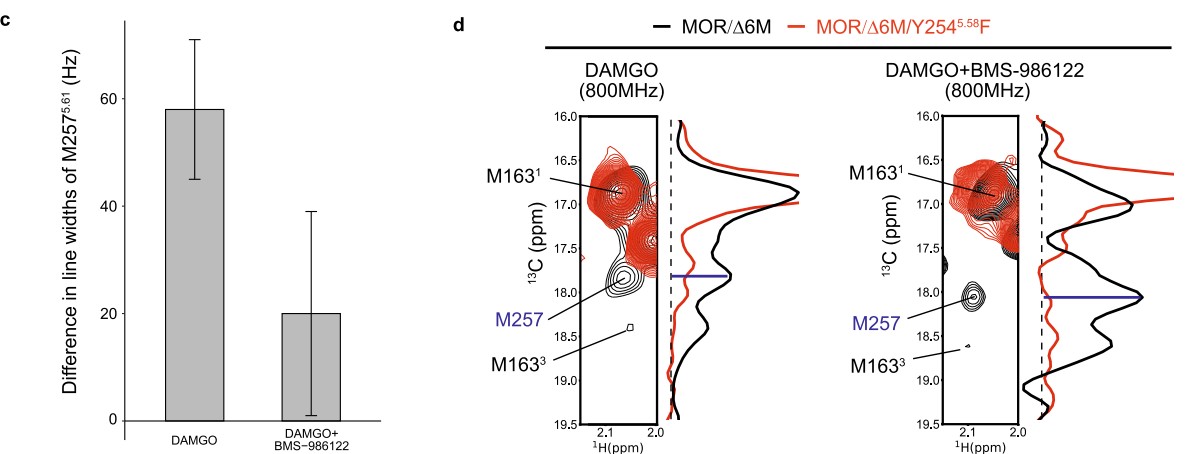

**Fig. 3 | M257$^{5.61}$ methyl NMR signal as a monitor for R167$^{3.50}$-Y254$^{5.58}$ interaction.**
**a** A closeup view of the cryo-EM structure of MOR(DAMGO) in complex with BMS-986122. The Cε atom of M257$^{5.61}$ is represented by a sphere. **b** $^1$H-$^{13}$C HMQC spectra of MOR(DAMGO) in the presence and absence of BMS-986122 at the $^1$H frequencies of 800 MHz and 1 GHz. Cross sections of the signal of M257$^{5.61}$ in the $^{13}$C direction are shown in black. Colored lines show the peaks obtained from spectral deconvolution of the cross sections. Linewidths of the M257 signal in the $^{13}$C direction are shown with blue highlights. **c** Differences in linewidths of the M257 signal between 800 MHz and 1 GHz. Error bars represent the standard errors from the line shape fitting to the 64 data points in the indirect dimension. Source data are provided as a Source Data file. **d** Overlays of $^1$H-$^{13}$C HMQC spectra and cross section in the $^{13}$C direction of the M257 signal of parental MOR/Δ6M (black) and the Y254$^{5.58}$F variant (red).

mutations destabilize the fully-activated conformation, which features the outward-shifted TM6.

Finally, the contribution of R167$^{3.50}$ and Y254$^{5.58}$ to the activity of the MOR was investigated using a GTP turnover assay[34–36] (Fig. 4c and Supplementary Fig. 8b). When normalized by the GTP turnover rate of the G$_i$ protein in the presence of MOR/Δ6M(DAMGO), the activities of the MOR/Δ6M/Y254$^{5.58}$F(DAMGO) variant were −4 ± 1% and 48 ± 7% in the absence and presence of BMS-986122 (Fig. 4c), and that of the MOR/Δ6M/R167$^{3.50}$L(DAMGO) variant were −17.3 ± 0.6% and −16.9 ± 1.6%, respectively (Supplementary Fig. 8b). The observation

that the activity was partly rescued by the addition of BMS-986122 in the Y254$^{5.58}$F variant indicated that the Y254$^{5.58}$F mutation does not abolish the structure of MOR nor the interaction with the G$_i$ protein. Therefore, these results demonstrated that the disruption of the interaction between R167$^{3.50}$ and Y254$^{5.58}$ drastically decreases the G$_i$ protein-stimulating activity of MOR, consistent with the decrease of the population of the fully-activated conformation observed in the NMR experiments (Fig. 4b and Supplementary Fig. 8a). This is further supported by the cellular cAMP inhibition assay, where substituting Y254$^{5.58}$ with phenylalanine abolished the activity of MOR

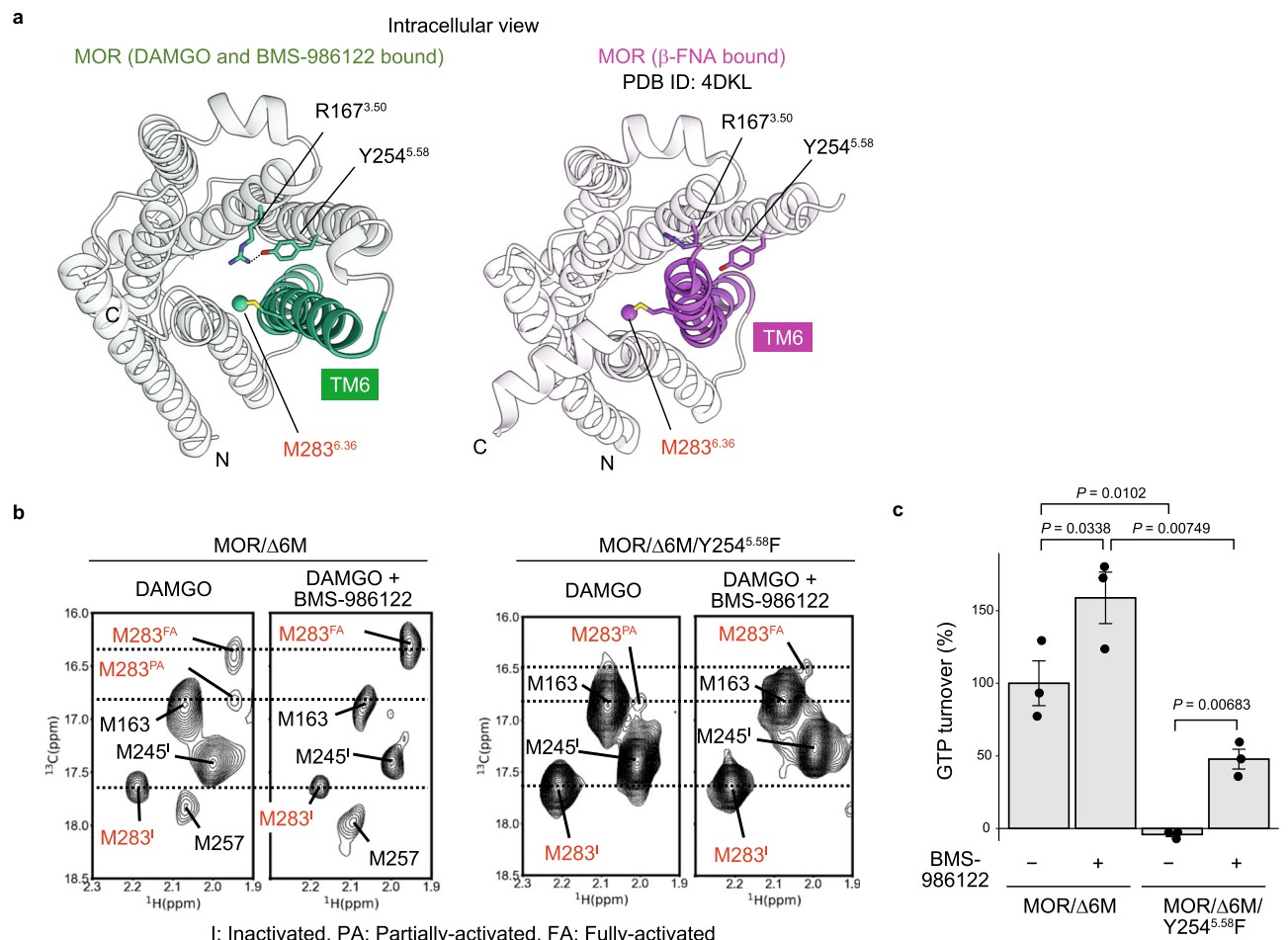

**Fig. 4 | Importance of R167[3.50]-Y254[5.58] hydrogen bonding interaction for function and structural dynamics of MOR. a** Intracellular view of structures of MOR bound to DAMGO and BMS-986122 (left) and bound to antagonist β-FNA (right, PDB ID: 4DKL). TM6 helices of MOR were colored green and magenta, and the Cε atoms of M283 were represented by spheres. **b** ¹H-¹³C HMQC spectra of the parental MOR/Δ6M (left) or the Y254[5.58]F variant (right) bound to DAMGO in the presence and absence of BMS-986122. **c** GTP turnover of the G$_i$ protein stimulated by MOR/Δ6M or the Y254[5.58]F variant in the DAMGO-bound state in the presence and absence of BMS-986122. All values were normalized by the GTP turnover rate of the G$_i$ protein in the presence of MOR/Δ6M in the full agonist DAMGO-bound state. Data are presented as mean ± standard error of the mean (s.e.m.) ($n$ = 3 independent replicates). Statistical significance was determined by a one-tailed Student's $t$ test, and $P$ values are indicated. Source data are provided as a Source Data file.

(Supplementary Fig. 9). Together, these results indicate that the interaction of R167[3.50] and Y254[5.58] is involved in the regulation of the function of MOR via the dynamic conformational equilibrium, and the binding of BMS-986122 enhances the function by shifting that conformational equilibrium.

## Discussion

In this study, we analyzed the three-dimensional structure of MOR(DAMGO)-G$_i$-scFv16 in the BMS-986122 bound state and showed that BMS-986122 binds to TM3, TM4, and TM5 from the lipid membrane side. The main chain traces of MOR were virtually identical between the cryo-EM structures in the absence and presence of BMS-986122. This similarity is likely because the G$_i$ protein alters the overall main chain conformation of MOR to the G$_i$ protein-bound form, largely canceling the structural and dynamic heterogeneity in the intracellular effector-unbound states, which regulate the function of GPCRs[1,19], as observed by the NMR analysis of MOR in the mini-G$_{s/i1}$ bound state (Supplementary Fig. 7). Although cryo-EM analyses of agonist-activated GPCRs in the absence of intracellular binders are generally difficult due to smaller molecular sizes and inherent function-related dynamics, comparison of the cryo-EM maps of MOR(DAMGO)-G$_i$-scFv16 in the absence and presence of BMS-986122 identified a

notable difference that may have remained after binding with the G$_i$ protein, as in the increased density between the conserved residues R167[3.50] and Y254[5.58]. A survey in the Protein Data Bank (PDB) database for class A GPCR structures in complex with their cognate G proteins or mini-G proteins revealed that in 106 out of 339 structures, distances between N in R[3.50] and O in Y[5.58] are closer than 3.0 Å, suggesting that the hydrogen bonding interaction between these residues is not specific to MOR[42] (Supplementary Fig. 10). Although this should not be the only structural difference caused by the binding of BMS-986122, our solution NMR analyses of MOR in the absence of G$_i$ proteins demonstrated that the conformational equilibrium around Y254[5.58] is suppressed by BMS-986122 and enhanced by the sole Y254[5.58]F mutation that removes the hydroxy group from Y254[5.58]. Concurrently, our NMR analyses revealed that the disruption of the hydrogen bonding interaction between R167[3.50] and Y254[5.58] decreases the population of the fully-activated conformation (Fig. 4b), where the intracellular half of TM6 in MOR is outward-shifted[19]. These suggest that the hydrogen bonding interaction between R167[3.50] and Y254[5.58] plays an important role in stabilizing the outward-shifted TM6, likely by blocking the inward shift of TM6 (Fig. 5). It should be stressed here that, in our hands, molecular dynamics simulations were not able to recapitulate the structural transition of the three conformations, possibly because

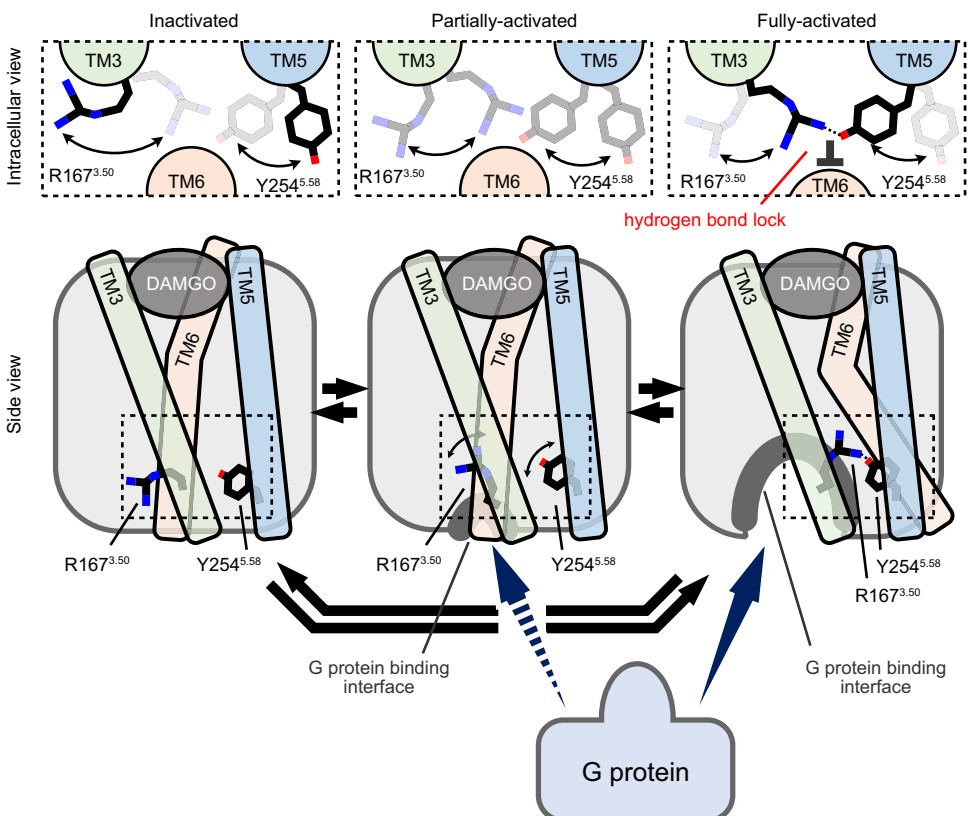

**Fig. 5 | The role of R167[3.50]-Y254[5.58] hydrogen bonding interaction in the function-related equilibrium of MOR.** A schematic model illustrating the effects of the hydrogen bonding interaction between R167[3.50] and Y254[5.58] on the conformational equilibrium of MOR is shown.

the process occurs on a timescale slower than 10 ms (estimated from the signal splitting in Hz) which is much slower than the window of simulation time (1 μs), thus making it difficult to study the allosteric modulation pathway by BMS-986122 in silico. This notion corroborates the importance of the experimental evidence of the dynamic properties of MOR by NMR. Together, the combined use of cryo-EM and NMR highlighted that the hydrogen bonding interaction between R167[3.50] and Y254[5.58] is crucial in the activation process of MOR. Binding of BMS-986122 may cause the stabilization of the hydrogen bonding interaction between R167[3.50] and Y254[5.58] for stronger activation of MOR than when bound to the full agonist DAMGO alone.

BMS-986122 binds to the outer surface in the transmembrane region on TM3, TM4, and TM5 and does not interact directly with the side chain atoms of R167[3.50] or Y254[5.58], which reside in the inner core region of MOR (Fig. 2). This suggests that the binding of BMS-986122 allosterically changes the local hydrogen bonding interaction between these two residues. We have previously reported that the binding of BMS-986122 destabilizes the partially-activated conformation through the hydrophobic cluster between TM3 and TM6, likely due to the rearrangement of TM3 by the interaction with BMS-986122[19]. In this study, we have demonstrated that the hydrogen bonding interaction between R167[3.50] and Y254[5.58] stabilizes the fully-activated conformation because the interaction may serve as a lock to stabilize TM6 in the outward-shifted position (Fig. 5). Since BMS-986122 binds to TM3 on the opposite side of R167[3.50], it is possible that the binding of BMS-986122 from the outer surface reorganizes the interhelical interaction of TM3, thus changing the relative distance between TM3 and TM5, which leads to the allosteric stabilization of the key hydrogen bonding interaction between the highly conserved R167[3.50] and Y254[5.58] in the core region. Considering the conservation of these residues (Supplementary Fig. 6), and the common observation of multiple states with varying activities across many GPCRs[19–25], the mechanism of allosteric

activation of MOR by BMS-986122, as reported herein, may reflect a broader activation paradigm in other GPCRs.

Previous structural analyses have showcased many allosteric modulator binding sites on various GPCRs[14]. Some of these studies have proposed a mechanism where the positive allosteric modulators stabilize the active state conformation (i.e., the conformation in the G protein-bound state) of GPCRs[43–46], but the structural basis of the stabilization remained uncharacterized, or different in each case. These imply that, compared to orthosteric ligands, there can be many structural pathways within GPCR molecules to allosterically modulate their activity. This observation makes it challenging to rationally design an allosteric modulator for a certain GPCR. The mechanism of the allosteric modulation of MOR by BMS-986122 shown here illustrates that the binding of the allosteric modulator laterally rearranges the interhelical interaction between the transmembrane helices, thus altering the key interaction between the conserved residues, R167[3.50] and Y254[5.58], at the core region. Therefore, modulation of this interaction laterally from the transmembrane region can be a promising target site to allosterically potentiate the activity of GPCRs more than the full agonist alone. This approach would be more straightforward than regulating the non-conserved hydrophobic interactions between TM3 and TM5[19].

In cells, the signaling of GPCRs is complexly controlled based on various factors including the expression levels of these receptors and the concentrations of endogenous agonists. This complexity makes it challenging to understand if there are evolutionary reasons why some GPCRs have evolved so that they do not elicit maximal activity solely via the binding of orthosteric ligands. However, it is noteworthy from a pharmaceutical standpoint that the concomitant use of allosteric modulators with orthosteric ligands can elicit higher activity in some GPCRs than would be expected from the efficacy of endogenous ligands alone. This is reflected in the NMR spectra of MOR in the full

agonist-bound state, where the population of the fully-activated conformation is not 100% (Fig. 4b). It may be possible to evaluate if allosteric modulation would be feasible for a certain GPCR for which allosteric modulators have not been identified, by observing the NMR spectra thereof in the full agonist-bound state. Identifying and controlling the function-related structural equilibria by analyzing NMR spectra, while complementing the structural information of the compound's binding site by cryo-EM analyses, presents an attractive strategy for the development of GPCR therapeutics, enabling the full activation of such GPCRs. We envision that the findings reported here may accelerate the development of analgesics targeting MOR, and therapeutics for diseases related to GPCRs.

## Methods

### Protein expression and purification for cryo-EM analyses

For cryo-EM analyses, a modified human MOR construct with a removable N-terminal Flag-tag (DYKDDDDA) and C-terminal octahistidine tag (HHHHHHHH)[29,30] was used in this study, with minor modifications. N-terminal residues (M1–G63) of MOR were replaced with the thermostabilized apocytochrome b562RIL from *Escherichia coli* (M7W, H102I, and R106L) (BRIL) protein and a linker sequence (GSPGARSAS). C-terminal residues (Q363-P400) of MOR were truncated. The N-terminal Flag-tag and C-terminal histidine tag were removable with rhinovirus 3C protease (TaKaRa 7360). The thermostabilizing mutation F158$^{3.41}$W, which does not interfere with the activity, was introduced to increase thermostability as well as the total expression level[31,32]. Sf9 cells (Gibco 11496015) in SF-II 900 SFM (Gibco 10902096) were infected with baculovirus at a density of $2 \times 10^6$ cells/mL and incubated for 48 h at 27 °C. The cell membrane was solubilized in n-dodecyl-β-D-maltoside (DDM, Nacalai Tesque 14239-54) and cholesterol hemisuccinate (CHS, Merck C6013) and purified by Ni-NTA resin (QIAGEN 30230). The eluate was further purified by M1 anti-Flag immunoaffinity resin (Millipore A4596) and size exclusion chromatography against 20 mM HEPES-NaOH (pH 7.5), 100 mM NaCl, 0.1% DDM, 0.01% CHS, 1 µM DAMGO (Cayman Chemicals, 21553). A further 100 µM of DAMGO was added to the peak fraction and then the peak fraction was concentrated to ~100 µM.

*Macaca mulatta* Gα$_{i3}$ subunit, N-terminally octahistidine-tagged human Gβ$_1$ subunit, and wild-type human Gγ$_2$ subunit were tandemly inserted into the pACEBac1 vector to enable coexpression of the three subunits. Sf9 cells (Gibco 11496015) in SF-II 900 SFM (Gibco 10902096) were infected with baculovirus at a density of $2 \times 10^6$ cells/mL and incubated for 48 h at 27 °C. The cells were collected and disrupted by nitrogen cavitation under 600 psi for 30 min. Lipid-modified heterotrimeric G$_i$ protein was extracted in a buffer containing 1% sodium cholate. The soluble fraction was purified using Ni-NTA chromatography (QIAGEN 30230) with the detergent exchanged from sodium cholate to DDM on a column. The eluate was dialyzed against 20 mM HEPES-NaOH (pH 7.5), 100 mM NaCl, 0.015% DDM, 100 mM TCEP, 10 mM GDP, and concentrated to ~10 mg/mL.

A C-terminally hexahistidine-tagged single-chain construct of Fab16 (scFv16) was cloned into the pFastBac vector containing a gp67 secretion signal prior to the N-terminus of scFv16. Sf9 cells (Gibco 11496015) in SF-II 900 SFM (Gibco 10902096) were infected with baculovirus at a density of $2 \times 10^6$ cells/mL and incubated for 48–72 h at 27 °C. The cells were spun down by centrifugation at 780×$g$ for 15 min, and the supernatant was concentrated about three-fold with a pressure-based sample concentration device (MWCO 10 kDa) (Merck UFSC40001). The concentrated supernatant was then dialyzed against a buffer containing 20 mM Tris-HCl (pH 8.0), 100 mM NaCl. The sample was then centrifuged at 2380 × $g$ for 30 min to remove the precipitant, and the supernatant was applied to Ni-NTA resin (QIAGEN 30230). After washing with 5 column volumes of buffer containing 20 mM Tris-HCl (pH 8.0) and 300 mM NaCl, 5 column volumes of buffer containing 20 mM Tris-HCl (pH 8.0), 100 mM NaCl, and 5 mM

imidazole, and 50 column volumes of buffer containing 20 mM Tris-HCl (pH 8.0), 100 mM NaCl, and 10 mM imidazole, the sample was eluted with buffer containing 20 mM Tris-HCl (pH 7.2), 100 mM NaCl, and 300 mM imidazole. The samples containing scFv16 were pooled, and 20 U/mg protein of HRV-3C protease (TaKaRa 7360) was added to cleave the hexahistidine-tag while dialyzing the sample against a buffer containing 20 mM Tris-HCl (pH 7.2) and 100 mM NaCl. The sample was then passed through Ni-NTA resin (QIAGEN 30230), concentrated, and purified by size exclusion chromatography using an ÄKTA system equipped with a Superdex 200 10/300 increase column (Cytiva 28990944). The fractions containing monomeric scFv16 were pooled, concentrated, flash-frozen in liquid nitrogen, and stored at −80 °C until use. All the plasmids used are available from the authors upon request.

### Assembly and purification of the MOR(DAMGO)-G$_i$-scFv16 complex

The formation of the MOR(DAMGO)-G$_i$-scFv16 complex was achieved according to the previous report[33]. The purified DAMGO-bound MOR was mixed with a 0.2 molar excess of heterotrimeric G$_i$ protein and then incubated for 1 h at 25 °C. Apyrase (New England Biolabs M0398S) was added to catalyze the hydrolysis of free GDP, followed by another incubation for 1 h at 25 °C. To exchange the detergent from DDM to lauryl maltose neopentyl glycol (LMNG, Anatrace NG310), a fourfold volume of 20 mM HEPES-NaOH (pH 7.5), 100 mM NaCl, 1% LMNG, and 0.1% CHS was added to the complex and incubated for an additional 1 h at 25 °C. 1 mM MnCl$_2$ and λ-phosphatase (New England Biolabs P0753S) were added to the complex and incubated for 2 h at 4 °C. The complex was purified using M1 anti-FLAG affinity chromatography (Millipore A4596) to remove residual DDM and excess heterotrimeric G$_i$ protein and the concentration of LMNG and CHS was gradually decreased to 0.01% and 0.001%, respectively. In total, 100 mM of TCEP and a 0.25 molar excess of scFv16 were added to the eluate and incubated overnight at 4 °C. MOR(DAMGO)-G$_i$-scFv16 was further purified by size exclusion chromatography on a Superdex 200 Increase 10/300 GL column (Cytiva 28990944) in 20 mM HEPES-NaOH (pH 7.5), 100 mM NaCl, 0.00075% LMNG, 0.00025% glycol-diosgenin (GDN, Anatrace GDN101), 0.0001% CHS, and 300 nM DAMGO. The peak fractions were supplemented with 100 µM BMS-986122 (Sigma-Aldrich SML0917) and incubated for 1 h at 4 °C. The peak fractions were then concentrated to approximately 5 mg/mL using Amicon Ultra-0.5 (MWCO 100 kDa) (Merck UFC510024) for electron microscopy studies.

### Cryo-EM sample preparation and image acquisition

Four µL of the purified MOR(DAMGO)-G$_i$-scFv16 complex, both with and without BMS-986122, were applied onto glow-discharged Quantifoil R1.2/1.3 300 mesh holey carbon grids using a Vitrobot Mark IV instrument (ThermoFisher Scientific). The application was performed with a blotting force set to 0 for 3 s at 100% humidity and 4 °C. After being plunge-frozen in liquid ethane, the grids were stored in liquid nitrogen and subjected to cryo-EM data collection and analysis.

### Cryo-EM data processing

Cryo-EM imaging was conducted using a Titan Krios G4 (ThermoFisher Scientific) operated at 300 kV, equipped with a Gatan Quantum-LS Energy Filter (slit width 15 eV) and a Gatan K3 direct electron detector at a nominal magnification of 105,000× in electron-counting mode, corresponding to a pixel size of 0.83 Å per pixel. Each set of movie stacks was recorded at a total of 15.5 electrons per pixel per second for 2.3 s, resulting in an accumulated exposure of 50.5 e$^-$ Å$^{-2}$. These data were automatically acquired by the image-shift method using EPU software with a defocus range of −0.8 to −2.0 mm. In the dataset of the MOR(DAMGO)-G$_i$-scFv16 complex, 6,450 movie stacks were acquired. All image processing was performed with RELION-3.1.3. Dose-fractionated image stacks were subjected to beam-induced motion correction using MotionCor2 and the contrast transfer function

parameters were estimated using CTFFIND version 4.1. A total of 8,500,757 particles were picked using crYOLO 1.9.7 from the micrographs and extracted at a pixel size of 1.65 Å. These particles were subjected to several rounds of 2D and 3D classifications. The selected 802,844 particles were then re-extracted at a pixel size of 0.83 Å and subjected to 3D refinement, Bayesian polishing, and subsequent postprocessing. The map's global resolution was improved to 2.98 Å, according to the Fourier shell correlation (FSC) = 0.143 criterion.

In the dataset containing BMS-986122, 5846 movie stacks were acquired. All image processing was performed with RELION-3.1.3. Dose-fractionated image stacks were subjected to beam-induced motion correction using MotionCor2 and the contrast transfer function parameters were estimated using CTFFIND version 4.1. A total of 3,452,964 particles were picked using Laplacian-of-Gaussian picking from the micrographs and extracted at a pixel size of 1.65 Å. These particles were subjected to several rounds of 2D and 3D classifications. The selected 308,249 particles were then re-extracted at a pixel size of 0.83 Å and subjected to 3D refinement, Bayesian polishing, and subsequent postprocessing of the map. The map's global resolution was improved to 3.05 Å, according to the Fourier shell correlation (FSC) = 0.143 criterion.

### Model building and refinement

A starting model of the MOR, Gi subunits, and scFv16 for the MOR(DAMGO)-$G_i$-scFv16 complex was based on PDB entries 7SBF and 6CRK. The models were docked into the cryo-EM map using UCSF ChimeraX[47] version 1.7, followed by manual adjustment in Coot 1.1 and refinement in phenix.real_space.refine[48] version 1.21. The refined model was used as the starting model for the MOR, Gi subunits and scFv16 for the data of MOR(DAMGO)-$G_i$-scFv16 containing BMS-986122. A model of BMS-986122 was generated by eLBOW[49] in Phenix[50] version 1.21. Visual inspection of the maps for the two datasets revealed continuous and elongated densities corresponding to BMS-986122 in the vicinity of Thr162 on the membrane-facing surface of TM3, 4, and 5 of MOR. Since the quality of the map was not sufficient for fitting the allosteric modulator model, possible conformations of the model were calculated using AutoDock Vina[51] version 1.2.5 through the UCSF Chimera[52] GUI. Nine poses were generated, of which the fourth model, with a score of −6.676, overlapping the elongated densities, was chosen as the modulator model with the most probable conformation.

### Expression and purification of MOR for NMR studies

For NMR analyses, a modified human MOR construct[19] with a removable N-terminal Flag-tag (DYKDDDDA) and a nonremovable C-terminal 8x histidine tag (HHHHHHHH), MOR/Δ6M was used. N-terminal residues (M1-A5) and C-terminal residues (Q363-P400) of MOR were truncated. A TEV protease recognition sequence (ENLYFQ) was introduced between G53 and G54. MOR/Δ6M or its variant was expressed and purified as previously described[4,19]. Sf9 cells (Gibco 11496015) were collected and resuspended in an amino-acid-deficient medium supplemented with non-labeled ADEQNGSPOHWC, deuterated FVYTLIKR, and [α, β, β, γ, γ-²H, methyl-¹³C] Met at a final concentration of $2 \times 10^6$ cells/mL. Then, these cells were infected with baculovirus and incubated for 48 h at 27 °C in the presence of 10 mM naloxone. The cell membrane was solubilized in DDM and CHS and purified by TALON resin (Thermo Scientific). The eluate was further purified and exchanged into a buffer composed of 20 mM HEPES-NaOH (pH 7.5), 100 mM NaCl, 0.01% LMNG, and 0.001% CHS on M1 anti-Flag immunoaffinity resin. After digestion of the N-terminal FLAG tag by TEV protease, MOR was further purified by size exclusion chromatography. Peak fractions were collected and exchanged into a buffer composed of 20 mM HEPES-NaOH (pH 7.5), 100 mM NaCl, 0.01% LMNG, 0.001% CHS, 100 μM DAMGO, and 250 mM

imidazole using TALON resin. Finally, the buffer was exchanged into a buffer composed of 20 mM sodium phosphate (pH 7.5) and 100 μM DAMGO in $D_2O$ using Amicon Ultra-4 (MWCO 30 kDa) (Merck UFC803024). All the plasmids used are available from the authors upon request.

### Preparation of mini-$G_{s/i1}$

The protein sequence for mini-$G_{s/i1}$ was derived from the preceding article[53]. The DNA sequence encoding mini-$G_{s/i1}$ with an N-terminal His-tag followed by a TEV protease cleavage sequence was inserted into the pET-43.1.a based vector (Novagen 70939). The *E. coli* strain BL21(DE3) (ThermoFisher Scientific C600003) was transformed with the resultant vector and cultured in LB medium containing 50 mg/L ampicillin. Protein expression was induced by the addition of isopropyl β-ᴅ-1-thiogalactopyranoside (IPTG) to a final concentration of 50 μM when the $OD_{600}$ reached 0.8. After 20 h of culture at 25 °C, cells were harvested by centrifugation at $9000 \times g$ for 15 min. The cells were resuspended in 100 mL of buffer containing 40 mM HEPES-NaOH (pH 7.5), 100 mM NaCl, 10% glycerol, 5 mM $MgCl_2$, 50 μM GDP, 28 μM E64, 10 μM leupeptin, 2.5 μM pepstatin A, 0.3 μM aprotinin, and 1 mM AEBSF. Cells were then disrupted by sonication and centrifuged at $38,000 \times g$ for 45 min at 4 °C. The supernatant was applied to His Select Nickel Affinity Gel (Millipore P6611) and washed with a buffer containing 20 mM HEPES-NaOH (pH 7.5), 500 mM NaCl, 10% glycerol, 1 mM MgCl2, 50 μM GDP, 1 mM DTT, and 40 mM imidazole. The His-tagged mini-$G_{s/i1}$ was then eluted from the column with a buffer containing 20 mM HEPES-NaOH (pH 7.5), 100 mM NaCl, 10% glycerol, 1 mM $MgCl_2$, 50 μM GDP, 1 mM DTT, and 500 mM imidazole. The fractions containing His-tagged mini-$G_{s/i1}$ were pooled and dialyzed overnight against a buffer containing 20 mM HEPES-NaOH (pH 7.5), 100 mM NaCl, 10% glycerol, 1 mM $MgCl_2$, 50 μM GDP, and 1 mM DTT. EDTA was added to a final concentration of 0.5 mM, and the His-tag was cleaved by TEV protease for 16 h at 4 °C. The sample was then passed through His Select Nickel Affinity Gel (Millipore P6611), and the flow-through fraction was concentrated using Amicon Ultra-15 (MWCO 10 kDa) (Merck UFC901024). The concentrated sample was then purified by size exclusion chromatography using a HiLoad 26/600 Superdex 75 pg (Cytiva 28989334). All the plasmids used are available from the authors upon request.

### NMR experiments

NMR experiments were conducted at 298 K in buffer containing 20 mM sodium phosphate (pH 7.5) and 100 μM DAMGO in $D_2O$, using Bruker Avance 800 and 900, or Ascend Evo 1.0 GHz spectrometers equipped with cryogenic probes. Typical sample concentrations are 10 μM. Spectra were processed and analyzed using TopSpin, version 3.5 or 4.0 (Bruker), and referenced to d6-sodium trimethylsilylpropanesulfonate (DSS, FUJIFILM Wako Chemicals 044-31671) as an external reference. Each MOR sample was recorded both in the absence and presence of 100 μM BMS-986122 with 1% (v/v) DMSO. The ¹H-¹³C HMQC spectra were acquired are reported previously[19] with the ¹H and ¹³C spectrum widths of 16.0 and 31.0 ppm, respectively. The typical number of scans are 256 with the number of complex data points in the indirect dimension of 128. Assignments of the methyl groups of methionine residues of MOR were transferred from our previous report[19], in which spectra were acquired under the identical NMR buffer and temperature. The linewidths of the signals in the ¹H-¹³C HMQC spectra were calculated by 2D deconvolution using the dcon function in TopSpin version 4.0.

### Guanosine triphosphate (GTP) turnover assay

The GTP turnover assay was carried out using a GTPase-Glo assay kit (Promega V7681). The purified DAMGO-bound MOR/Δ6M was used for the assay. After incubation for 1 h in the presence of 10 mM BMS-986122 or vehicle (1% (v/v) dimethyl sulfoxide (DMSO)) on ice, the

GDP–GTP exchange reaction was initiated by mixing purified MOR in LMNG micelles with the heterotrimeric $G_i$ protein at final concentrations of 0.2 μM and 0.5 μM, respectively, in an assay buffer containing 20 mM sodium phosphate (pH 7.5), 0.01% LMNG, 0.001% CHS, 10 mM $MgCl_2$, 1 μM GDP, 5 μM GTP, and 100 μM DAMGO. After incubation for 30 min at 25 °C, the same volume of reconstituted GTPase-Glo reagent was added to the sample and incubated for another 30 min at 25 °C. Luminescence was measured after the addition of the detection reagent and a 5-min incubation at 25 °C, using an EnVision 2105 multi-mode plate reader (PerkinElmer). The relative turnover rate of each state of MOR was calculated as the ratio of the corrected response to that of MOR/Δ6M bound to DAMGO in the absence of BMS-986122.

## cAMP inhibition assay

To measure $G_i$-mediated cAMP inhibition, HEK293T cells (KAC Co.,Ltd. EC12022001-F0) were co-transfected with wild-type MOR/Δ6M (with the N-terminal sequence that was cleaved by TEV protease in the NMR experiments) or the Y254[5.58]F variant, along with a luciferase-based cAMP biosensor (GloSensor, Promega)[54]. In total, 1.8 μg of pGloSensor-22F cAMP plasmid (Promega E2301) and 0.2 μg of the pCI-neo vector (Promega E1841) harboring wild-type MOR/Δ6M, the Y254[5.58]F variant of MOR/Δ6M, or the empty backbone pCI-neo vector, were mixed and transfected into $5 \times 10^5$ HEK293T cells cultured in a six-well plate for 24 h in DMEM (Nacalai Tesque 08458-16) supplemented with 1.0% dialyzed FBS (ThermoFisher Scientific 26140079), using X-tremeGENE 360 (Roche 8724105001). After 24 h, cells were harvested and resuspended into a 96-well white flat-bottom tissue-culture-treated microplate at $2.0 \times 10^4$ cells/well, in $CO_2$-independent medium (Gibco 18045088) supplemented with 1% (v/v) dialyzed FBS and 2% (v/v) GloSensor cAMP Reagent (Promega E1290). After incubating for 2 h at room temperature, DAMGO and/or BMS-986122 were added to final concentrations of 1 μM, and the plates were further incubated for 15 min at room temperature. Then, isoproterenol (Sigma-Aldrich I6504) was added to a final concentration of 1 μM to activate $G_s$ protein via endogenous $\beta_2$-adrenergic receptors, and luminescence intensity was quantified 15 min later with an EnVision 2104 multilabel plate reader (PerkinElmer) for 10 s. The degree of $G_i$ activation was quantified by subtracting the luminescence values from control wells, where buffer was added instead of DAMGO or BMS-986122. The experiments were repeated six times, and mean values and standard errors are reported. All the plasmids used are available from the authors upon request.

## Reporting summary

Further information on research design is available in the Nature Portfolio Reporting Summary linked to this article.

## Data availability

The cryo-EM density map data generated in this study have been deposited in the Electron Microscopy Data Bank (EMDB) database under accession codes EMD-36990 (MOR(DAMGO)-$G_i$-scFv16 complex in the presence of BMS-986122) and EMD-36989 (that in the absence of BMS-986122). The atomic coordinate data generated in this study have been deposited in the PDB database under accession code 8K9L (MOR(DAMGO)-$G_i$-scFv16 complex in the presence of BMS-986122) and 8K9K (that in the absence of BMS-986122). Other structure data used in this study are available in PDB under accession codes 7SBF, 4DKL, 6CRK, 6DDE, and 6DDF.  Source data are provided with this paper.

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

## Acknowledgements

This work was supported by JSPS KAKENHI Grant Numbers JP17H06097 (to I.S.), JP21H02619 (to S.I.), and JP23KJ0574 (to S.K.). This work was also supported by grants from the Japan Agency for Medical Research and Development (AMED) Grant Number JP21ae0121028 (to I.S.), and RIKEN BDR Structural Cell Biology Project (to I.S. and M.S.). The cryo-EM experiments were performed at the RIKEN Yokohama cryo-EM facility.

## Author contributions

S.K., S.I. and I.S. designed the research. S.K., S.I. and N.A. prepared the protein samples. S.K., S.I. and N.A. conducted NMR experiments and analyzed the NMR data. T.U.-K., T.H. and M.S. conducted cryo-EM experiments and analyzed the cryo-EM data. S.K., S.I. and I.S. wrote the manuscript.

## Competing interests

The authors declare no competing interests.
