## [Peer Review File · Nature Communications]

Structural and dynamic insights into the activation of the μ -opioid receptor by an allosteric modulatorEditorial Note: Parts of this Peer Review File have been redacted as indicated to remove third-party material where no permission to publish could be obtained.

REVIEWER COMMENTS

Reviewer #1 (Remarks to the Author):

The authors explore the effects of a previously discovered positive allosteric modulator on the active state structure of an opioid-Gi complex. Their work reveals a signatory contact between R1673.50 and Y2545.58 which they propose is critical to enhancing G protein coupling (through nucleotide hydrolysis measurements). As the cryo-EM structures with and without the allosteric drug BMS-986122 revealed very little difference, additional mutagenesis and NMR was used to confirm the binding site of the drug and the stabilization of the active state. While interesting, I respectfully feel that the paper is an incremental improvement over their recently published and exceptional article (PNAS, 2022, Kaneko et al) where they identify a correlation between each of the two activation states observed by NMR and the role of the allosteric modulator to enhance the more active state. In their original paper, line broadening and mutagenesis identified the identical hydrophobic groove in which the allosteric modulator sits. In this paper, they point out that the R167-Y254 interaction in this groove is likely the key to enhancement of efficacy, while validating with additional (and convincing) line broadening measurements with/without a non-functional analog of BMS-986122. I have several comments/suggestions below

1) I can fully understand the intention of the authors but I believe it needs a critical revision by an expert English writer. Several paragraphs begin with "However", the title lacks an indefinite article, and many sentences are grammatically awkward "e.g. the introduction of point mutations, not to the residues that directly interact". Other small issues that need qualification or rewriting: " at a 3.1 Å resolution", "supporting that the ?? density originates from", " we observed a density on the membrane-facing surface", " NMR signal from M245 was largely perturbed: It was broadened" (semicolon versus colon), " This funding indicates that the...", GTP turnover assay? (Technically it is a GTP-ase assay which measures nucleotide exchange and hydrolysis), " The elute was further purified", Hepes versus HEPES.

2) The authors write "Our results also revealed that BMS-986122 increases the population of a conformation with an outward-shifted TM6, referred to as a fully-activated conformation in the three-state conformational equilibrium, thereby enhancing MOR activity. We have also shown that the hydrophobic interactions between TM3 and TM5 are involved in the destabilization of the partially-activated conformation, a conformation where TM6 is inward-shifted."

I personally think that the authors take liberties with the term "fully activated". G protein coupling and nucleotide exchange should involve a dynamic sequence of steps (associated with corresponding spectroscopic states) in which the G protein is first recognized and bound, then dynamically engaged so as to achieve nucleotide exchange, whereupon GTP binds and the alpha subunit dissociates. With this in mind, what is the fully active state? One can imagine unique TM6 activation intermediate states associated with binding GDP-bound Gi (the cryo-EM structure) and the subsequent removal of nucleotide, which from a pharmacological perspective might be described as fully active. The NMR however, is not performed in the presence of Gi nor in the presence of mini-Gi, so the exact role of the TM6 active resonances (1 and 2) isn't clear. It is also not impossible that the so-called "partially active

state" is on pathway to the fully active state. I don't think the authors need to solve this problem but I do think that it merits discussion and they need to be more cautious with regard to labeling a peak as "fully active", without any definition or in the absence of NMR with mini-Gi, which I would recommend if they wish to connect with their cryo-EM work. It's also a shame that the paper and the previous PNAS paper doesn't reference other NMR work that has actually dealt with multiple activation states in the presence and absence of G protein. For example, Huang et al, Cell, 2021 reveals three activation states, two of which are associated with partial and full agonism, and have fully qualified the meaning of active states and activation intermediates, with regard to G protein coupling. The presence of two activation states, one of which is associated with lower efficacy, seems to be quite common and many believe there are general reasons for this.

3) The authors write " However, the structural basis underlying the activity enhancement by the allosteric modulators is still largely unknown, hampering the rational design of allosteric modulators." This is firstly, a sweeping statement. One can find many papers that address allostery through MD, Monte Carlo analysis, and computational methods that hone in on allostery. Secondly, if the goal is to identify R167-Y254 as a key activation switch associated with allostery, would it not make sense to invoke any of the above computational methods to understand its dynamic and hence, allosteric role?

4) " The mainchain traces of MOR were virtually identical between the cryo-EM structures in the absence and presence of BMS-986122. This similarity is probably because the Gi protein alters the overall mainchain conformation of MOR to the Gi protein bound form, canceling the structural and dynamic differences in the intracellular effector-unbound states, which regulate the function of GPCRs".

I suggest a more cautious conclusion. The cryo-EM structure captures a single GDP-bound state. It's absolutely likely that dynamic equilibria are involved. Ideally, you'd need to extend the NMR observations to include mini-Gi to identify relative differences of functional states during G protein coupling, or capture cryo-EM states without nucleotide.

5) "Considering the conservation of these residues (Supplementary Fig. 5), the activation mechanism of MOR reported here might underlie the activation processes in many GPCRs."

I suggest the authors evaluate active state complexes of various class A receptors with their cognate G proteins and show if there is a prevalent R167-Y254 hydrogen bond or not. Both residues are after all important in polar networks and the NPXXY motif.

6) Presumably pharmacologists would ask why full efficacy hasn't evolved for the endogenous opioid receptor. Could it be that the BMS-activated state leads to greater desensitization over time? Is the BMS-activated state more prone to phosphorylation? Or might it be that in vivo, there are other adjuvants that rescue high efficacy signalling? The authors could nicely round out this paper if they were to add a paragraph or two to the discussion, bringing in the pharmacology of BMS-986122 in light of their observations.

Reviewer #2 (Remarks to the Author):

In this study, the authors present the cryo-EM structure of a DAMGO (full agonist) – MOR (μ -opioid receptor) – Gi – BMS-986122 (allosteric modulator) complex. They utilized NMR data to ascertain the orientation of BMS-986122 within this structure. Comparing the MOR–Gi structures with and without BMS-986122, a connecting density between residues R167(3.50) and Y254(5.58) was observed only in the BMS-986122-bound state. Further, NMR was employed to track conformational shifts in M257, near Y254, and M283 at the cytoplasmic end of TM6, revealing distinct MOR conformations modulated by BMS-986122.

While the exploration of the molecular mechanism of allosteric modulators in GPCRs is compelling, the conclusions drawn are not sufficiently supported by the data presented.

Major Comments:

1. The authors suggest a transient interaction between R167(3.50) and Y254(5.58) is enhanced by BMS-986122, contributing to MOR activation. However, this interaction is one among many in the active state of class A GPCRs. Without examining all interactions stabilizing MOR's active state, attributing BMS-986122's effect solely to this interaction is speculative. A more cautious conclusion might be that BMS-986122 stabilizes the active state conformation, akin to other positive allosteric modulators (e.g., PMID: 34497422).
2. The GTP turnover assay indicates that the Y254F mutant retains approximately 48% activity with BMS-986122, suggesting the R167-Y254 interaction is not solely responsible for BMS-986122's efficacy.
3. Additional cryo-EM data is needed to increase the resolution and confirm BMS-986122's conformation in the structure.

Minor Comments:

1. Page 3: Replace “extracellular active site” with “ligand binding site” for accuracy.
2. Figure 1: Include the density for DAMGO, as was done for BMS-986122, for comparative purposes.
3. The manuscript would benefit from ligand binding data to substantiate claims regarding F180 and W158's interaction with BMS-986122.
4. Figure 2B: Given the ~ 3 Å resolution, the significance of differences in density maps is unclear, especially as side chains appear similarly oriented.
5. The manuscript contains numerous speculative statements, such as the H-bond interaction between R167 and Y254 stabilizing the outward-shifted TM6 in the absence of G protein, without adequate experimental evidence.

Reviewer #3 (Remarks to the Author):

The manuscript presents an insightful study on the three-dimensional structure of the μ -opioid receptor

(MOR) in complex with the Gi protein and an allosteric modulator, BMS-986122, using cryogenic electron microscopy (cryo-EM). The authors indicate that binding of BMS-986122 induces alterations in map densities associated with R167^{3.50} and Y254^{5.58}, pivotal residues within conserved structural motifs among class A G-protein-coupled receptors (GPCRs). The findings provide valuable insights into the mechanisms underlying the variations in signaling activity by allosteric modulators. This manuscript is generally well written, the figures are of relatively good quality, and the conclusions are supported by the data. However, there are a few points that should be addressed prior to its acceptance for publication, including rewriting the manuscript and performing some additional experiments, as follows:

1. The NMR analyses complement the structural findings and add depth to the study. The enhancement of the interaction between R167^{3.50} and Y254^{5.58} by BMS-986122, thus stabilizing the fully-activated conformation is a crucial observation. However, NMR experiments were conducted in the absence of G proteins, which differs from the physiological conditions. Therefore, I suggest supplementing the conclusions with additional molecular and cellular-level functional experiments, such as cAMP accumulation assays, NanoBiT experiments, or any other pertinent methodology of the authors' choice. This would serve to strengthen the aforementioned conclusions.
2. Lines 125 to 127: Given that the structure has already been solved before, why did the authors determine it again? Are the two structures identical or any discernible differences? Please include additional descriptive elements to illustrate.
3. Lines 142 to 145: The structures of BMS-986124 and BMS-986122 exhibit remarkable similarity, yet only BMS-986122 acts as a positive allosteric modulator for MOR. Can authors speculate on the potential reasons for this phenomenon based on the current structural information available?
4. In the provided "wwPDB validation report" files, the second section "Entry composition" displays that the amino acid at position 158 in the R chain is modeled as Trp, while it is actually Phe. I did not find any mention of this mutation in the manuscript. Please provide clarification or include a statement addressing this discrepancy in the manuscript.
5. Some abbreviations are not explained when they first appear in the text, for example, line 98, "cryo-EM" and line 145, "HMQC" are not explained.
6. The numbering of amino acids lacks consistency. Please standardize it according to the Ballesteros-Weinstein numbering system. For instance, line 181, R167 should be revised to R167^{3.50}.
7. In Figure 1, panel A, please provide threshold level for the density under the current view, and please check all figures in the manuscript.
8. In Figure 3, Please maintain uniformity in the formatting of the y-axis labels for both the B panel and D panel.
9. Line 341, "MWCO 10K" should be revised as "MWCO 10KDa"
10. Line 433, "PHHHHHHHH" should be revised as "HHHHHHHHH"
11. Please ensure consistency in the citation format, for instance, line 541, DOI is not required.

Point-to-point responses to the reviewer comments

We would like to express our sincerest gratitude to all the referees for their valuable comments and suggestions on our manuscript. In response to their insightful feedback, we have meticulously revised the manuscript. Below, we provide our point-by-point responses to the reviewers' comments. The comments from the reviewers are displayed in blue font, whereas the revisions made to the manuscript are highlighted in red font. The page and line numbers correspond to those in the source DOCX file. We are confident that these modifications have satisfactorily addressed all concerns raised by the reviewers.

Reviewer #1:

The authors explore the effects of a previously discovered positive allosteric modulator on the active state structure of an opioid-Gi complex. Their work reveals a signatory contact between R167^{3,50} and Y254^{5,58} which they propose is critical to enhancing G protein coupling (through nucleotide hydrolysis measurements). As the cryo-EM structures with and without the allosteric drug BMS-986122 revealed very little difference, additional mutagenesis and NMR was used to confirm the binding site of the drug and the stabilization of the active state. While interesting, I respectfully feel that the paper is an incremental improvement over their recently published and exceptional article (PNAS, 2022, Kaneko et al) where they identify a correlation between each of the two activation states observed by NMR and the role of the allosteric modulator to enhance the more active state. In their original paper, line broadening and mutagenesis identified the identical hydrophobic groove in which the allosteric modulator sits. In this paper, they point out that the R167-Y254 interaction in

this groove is likely the key to enhancement of efficacy, while validating with additional (and convincing) line broadening measurements with/without a non-functional analog of BMS-986122. I have several comments/suggestions below

We would like to express our gratitude to the reviewer for their comprehensive review of our manuscript. We value the constructive feedback and have meticulously revised the manuscript following the reviewer's insightful suggestions. All comments and recommendations have been thoroughly addressed. We believe that these revisions have enhanced the manuscript, making it suitable for publication in *Nature Communications*.

Reviewer#1-1a) I can fully understand the intention of the authors but I believe it needs a critical revision by an expert English writer.

We appreciate the reviewer's comment regarding the need for a critical revision by an expert English writer and apologize for the grammatical errors in the original manuscript. In accordance with the reviewer's suggestion, we have employed an English proofreading service to review and correct all sentences in the current version of the manuscript.

Reviewer#1-1b) Several paragraphs begin with "However",

We appreciate the reviewer's suggestion and have modified the manuscript to ensure that no paragraph starts with 'However'.

Results (P.9, lines 149-154):

The position of this additional density is consistent with our previous result that the substitution of T162^{3,45} on TM3 to a methionine abrogated the binding of BMS-986122¹⁹, supporting the notion that the aforementioned additional density originates from BMS-986122 bound to MOR (Fig. 1b). **However**, due to the relatively weak EM density at this allosteric site, as well as the moderate resolution of 3.1 Å, it was not possible to unambiguously model BMS-986122 solely by the cryo-EM data.

Reviewer#1-1c) the title lacks an indefinite article,

We appreciate the reviewer's suggestion and have changed the title to include an indefinite article, now reading 'Structural and Dynamic Insights into the Activation of the μ -Opioid Receptor by **an** Allosteric Modulator'.

Reviewer#1-1d) and many sentences are grammatically awkward "e.g. the introduction of point mutations, **not to the residues that directly interact**".

We appreciate the reviewer's suggestion and have revised the sentence to 'For example, the introduction of point mutations, **not at residues** that directly interact with the intracellular effectors, ...' (P.3, lines 52-54). All other sentences have also been reviewed and corrected by an English proofreading service.

Reviewer#1-1e) Other small issues that need qualification or rewriting: " at a 3.1 Å resolution", "supporting that the ?? density originates from", " we observed a density on the membrane-facing surface", " NMR signal from M245 was largely perturbed: It was broadened" (semicolon versus colon), " This funding indicates that the...",

We thank the reviewer for pointing out these mistakes, and have corrected them accordingly. 'At a 3.1 Å resolution' has been corrected to 'at 3.1 Å resolution' (P.8, line 135). 'Supporting that the density originates from' has been revised to 'supporting the notion that the aforementioned additional density originates from' (P.9, lines 151-152). 'We observed a density on the membrane-facing surface' has been updated to 'we observed density on the membrane-facing surface' (P.9, line 146). 'NMR signal from M245 was largely perturbed: It was broadened...' has been corrected to 'NMR signal from M245^{5,49} was largely perturbed; it was broadened' (P.9, line 162 – P.10, line 163). 'This funding indicates that the...' has been corrected to 'these findings indicate that the...' (P.15, line 264).

Reviewer#1-1f) GTP turnover assay? (Technically it is a GTP-ase assay which measures nucleotide exchange and hydrolysis),

Yes, what we conducted was indeed a GTP-ase assay, which measures nucleotide exchange and hydrolysis, as the reviewer correctly noted. This assay is employed to measure the activity of GPCRs by observing the amounts of GTP consumed per unit time through GTP-GDP turnover by G proteins. The term 'GTP turnover assay' is widely utilized in numerous preceding articles, for example, Strohmman *et al.*, *Nat. Commun.* (2019) 10:2234, Chen *et al.*, *Nat. Commun.* (2022) 13:2375, and

Kumar *et al.*, *Nat. Commun.* (2023) 14:2672. We have amended the manuscript to briefly explain the assay with citations to these articles, thereby providing an introduction to the assay for our readership.

Results (P.11, lines 182 – 188):

This is further confirmed by a GTP turnover assay that measures the consumption of GTP by G proteins through GTP to GDP turnover^{34–36} (Supplementary Fig. 5c). F180^{ICL2}L exhibited a loss of BMS-986122-dependent increase in the activity of MOR, while W158^{3.41}M exhibited an increase in the activity upon the addition of BMS-986122 as the parental MOR/ Δ 6M, suggesting that F180^{ICL2} is more important for the allosteric modulation activity of BMS-986122 than aromatic residues at 3.41.

References

34. Strohman, M. J. *et al.* Local membrane charge regulates β 2 adrenergic receptor coupling to Gi3. *Nat Commun* **10**, 2234 (2019).
35. Chen, G. *et al.* Activation and allosteric regulation of the orphan GPR88-Gi1 signaling complex. *Nat Commun* **13**, 2375 (2022).
36. Krishna Kumar, K. *et al.* Structural basis for activation of CB1 by an endocannabinoid analog. *Nat Commun* **14**, 2672 (2023).

Reviewer#1-1g " The elute was further purified", Hepes versus HEPES.

“The elute was further purified” has been corrected to ‘The eluate was further purified’ (P.22, line 395 and P.29, line 518). ‘Hepes’ has been corrected to ‘HEPES’ throughout the manuscript.

Reviewer#1-2a) The authors write "Our results also revealed that BMS-986122 increases the population of a conformation with an outward-shifted TM6, referred to as a fully-activated conformation in the three-state conformational equilibrium, thereby enhancing MOR activity. We have also shown that the hydrophobic interactions between TM3 and TM5 are involved in the destabilization of the partially-activated conformation, a conformation where TM6 is inward-shifted." I personally think that the authors take liberties with the term "fully activated". G protein coupling and nucleotide exchange should involve a dynamic sequence of steps (associated with corresponding spectroscopic states) in which the G protein is first recognized and bound, then dynamically engaged so as to achieve nucleotide exchange, whereupon GTP binds and the alpha subunit dissociates. With this in mind, what is the fully active state?

We appreciate the reviewer's critical perspective on the use of the term 'fully activated'. To clarify, we did not utilize the term 'fully active state' throughout the manuscript. From the standpoint of GDP-GTP enzymatic exchange activity, fully active states should be those where GPCR and G protein form complexes, enabling G proteins to achieve nucleotide exchange and bind to GTP for hydrolysis, as the reviewer pointed out. The fully-activated state of MOR in our manuscript is defined as the state with the highest efficiency regarding coupling with the G protein, existing under structural equilibrium with other states (inactivated and partially-activated), in the absence of G protein. Here the focus is on MOR's ability to stimulate the G protein's enzymatic activity. In our preceding paper (Kaneko *et al.*, *Proc Natl Acad Sci USA* (2022) **119**:e2121918119), we showed this state is structurally characterized by an outward-shifted (i.e., open) TM6, enabling it to readily engage with G proteins. These states exist in the absence of intracellular effectors, predefining the apparent level of activity to interact with these effectors. Thus, what we observe by NMR is the

dynamic state of MOR prior to G protein coupling and nucleotide exchange.

Reviewer#1-2b) One can imagine unique TM6 activation intermediate states associated with binding GDP-bound G_i (the cryo-EM structure) and the subsequent removal of nucleotide, which from a pharmacological perspective might be described as fully active.

As written above in the response to Reviewer#1-2a), we agree with the reviewer's point that TM6 activation intermediate states associated with the G_i protein in the GDP-bound state can be described as 'fully active' with respect to the GTPase enzymatic activity of the G protein. Recognizing the precedence and widely accepted terminology, we have therefore avoided using the term 'fully active' to describe the states of MOR in the absence of G proteins. Instead, we have used the term 'fully-activated' to focus on the activity of MOR, which is to stimulate the GTPase activity of G proteins. The fully-activated conformation of MOR, characterized by the outward-shifted TM6 as demonstrated by our previous NMR analyses, resembles active conformations observed in previous three-dimensional structures of GPCRs in complex with G proteins, where the TM6 helix is outward-shifted.

Reviewer#1-2c) The NMR however, is not performed in the presence of G_i nor in the presence of mini- G_i , so the exact role of the TM6 active resonances (1 and 2) isn't clear.

In our preceding article (Kaneko *et al.*, *Proc Natl Acad Sci USA* (2022) **119**:e2121918119), we demonstrated that the activity of MOR correlates with the populations of three structural states

existing in the absence of G proteins. This implies that the efficiency of MOR's coupling with G proteins is determined prior to its interaction with them. Specifically, the intensity of resonance 1, as shown in the aforementioned article (shown below for your convenience), is correlated with the population of a structural state of MOR that exhibits the highest activity in stimulating the G protein. Conversely, the intensity of resonance 2 correlates with the population of a structural state exhibiting intermediate activity. Structurally, the distinctions between these two states lie in the positions of the intracellular tips of TM6, which are outward- and inward-shifted in states 1 and 2, respectively. The exact roles of these states, as reflected by these resonances, are to predetermine the apparent activity of MOR by their altered populations depending on various conditions. In essence, the effectiveness with which G proteins are stimulated by MOR is predetermined in the absence of G proteins, as it hinges on the probability of MOR assuming a particular structural state upon encountering a G protein.

[REDACTED]

Fig. 1 from Kaneko *et al.*, *Proc Natl Acad Sci USA* (2022) 119:e2121918119

Reviewer#1-2d) It is also not impossible that the so-called "partially active state" is on pathway to the fully active state.

Indeed, as the reviewer suggests, it is conceivable that the partially activated state of MOR in the absence of G proteins, as reflected by resonance 2, could be on the pathway to the 'fully active' state, which corresponds to the MOR structure in the G protein-bound state. In such a scenario, the TM6 helix would open following initial interaction with a G protein. This process might be less efficient compared to situations where G proteins directly interact with the fully-activated state, in which the TM6 helix is already open.

Reviewer#1-2e) I don't think the authors need to solve this problem but I do think that it merits discussion and they need to be more cautious with regard to labeling a peak as "fully active", without any definition or in the absence of NMR with mini-Gi, which I would recommend if they wish to connect with their cryo-EM work.

We appreciate the reviewer's constructive feedback regarding the terminology used to describe the states of MOR in the absence of G proteins and apologize for any confusion caused. As has been written above (Reviewer#1-2 a, b, c, and d), the terms 'fully-activated', 'partially-activated', and 'inactivated' states in our manuscript are used exclusively to describe the states of MOR in the absence of G proteins which are under equilibrium. In accordance with the reviewer's suggestion,

we have defined these terms in the paragraph where these words first appear in the main text, as follows:

Introduction (P.5, line 83 – P.6, line 95):

In preceding studies^{4,19}, we used nuclear magnetic resonance (NMR) spectroscopy to uncover a three-state conformational equilibrium at the intracellular half of the transmembrane helix 6 (TM6) of MOR, in the absence of intracellular binders. This conformational equilibrium controls the activity of MOR, as shown by the correlation of the populations of the three interchanging conformations with the activity. From this correlation, we designated these conformations, possessing high, intermediate, and no relative activities, as the fully-activated, partially-activated, and inactivated conformations, respectively¹⁹. In the absence of G proteins, these conformations are under equilibrium, where the populations are changed depending on the efficacies of the ligand bound to the extracellular ligand binding pocket. Furthermore, we also demonstrated that the intracellular half of TM6 is shifted outward in the fully-activated conformation in the absence of G proteins as in the GPCR structures bound to their cognate G proteins¹⁹.

References

4. Okude, J. *et al.* Identification of a Conformational Equilibrium That Determines the Efficacy and Functional Selectivity of the μ -Opioid Receptor. *Angew Chem Int Ed Engl* **54**, 15771–6 (2015).
19. Kaneko, S. *et al.* Activation mechanism of the μ -opioid receptor by an allosteric modulator. *Proc Natl Acad Sci U S A* **119**, e2121918119 (2022).

Furthermore, according to the reviewer's suggestion, we have acquired the NMR spectrum of MOR in the mini- $G_{s/i1}$ bound state (Supplementary Fig. 7a). mini- $G_{s/i1}$ is a chimera of mini- G_s and the C-terminal H5 helix of G_{i1} that specifically binds to G_i -coupled GPCRs with more thermostability than mini- G_i (Nehmé, *PLoS One* (2017) 12:e0175642). In the mini- $G_{s/i1}$ bound state, the NMR signal of M283 exhibited only one signal, which is different from all three signals observed in the absence of mini- $G_{s/i1}$. Since M283 is on the binding interface of the G_i protein (Supplementary Fig. 7b), this does not mean that none of the three signals are 'active', but indicates that the signal reflects the local environment of M283 in the mini- $G_{s/i1}$ -bound state. The fact that only one signal was observed in the mini- $G_{s/i1}$ -bound state clearly illustrates that the local conformation of TM6 is converged into a single conformation in an NMR chemical shift timescale (slower than μ s in this case), which should be like the G_i protein-bound state observed in the cryo-EM structure. Although it is of course possible that dynamics and/or structural heterogeneities exist in different parts of MOR, these results clearly show that the three-state exchanging dynamics of MOR at the intracellular tips of TM6 observed in the absence of the intracellular effectors are suppressed by the binding of mini- $G_{s/i1}$.

Supplementary Fig. 7 | ^1H - ^{13}C HSQC spectra of MOR/ $\Delta 6\text{M}$ (DAMGO) in complex with mini- $\text{G}_{\text{s/i1}}$

a ^1H - ^{13}C HMQC spectra of MOR/ $\Delta 6\text{M}$ (DAMGO). Black: MOR/ $\Delta 6\text{M}$ in the full agonist DAMGO-bound state, red: MOR/ $\Delta 6\text{M}$ in the full agonist DAMGO-bound state in complex with the engineered G_i protein, mini- $\text{G}_{\text{s/i1}}$. In the mini- $\text{G}_{\text{s/i1}}$ bound state, only one signal was observed from M283^{6,36}, indicating that the structural equilibrium among the fully-activated (FA), partially-activated (PA), and the inactivated (I) conformations is suppressed by the binding of mini- $\text{G}_{\text{s/i1}}$.

b A closeup view of the cryo-EM structure of MOR(DAMGO)- G_i -scFv16 (PDB ID: 6DDF). The C ϵ atom of M283^{6,36} is shown as a violet sphere, which is on the interface with the $\text{G}\alpha_i$ in the trimeric G_i protein, indicating that the chemical shift of the methyl group would be directly affected by the binding of the G_i protein or mini- $\text{G}_{\text{s/i1}}$.

We are thankful that the reviewer raised this point, and have added the NMR spectrum of MOR in the mini- $\text{G}_{\text{s/i1}}$ bound state in Supplementary Fig. 7, and mentioned these data in the main text, as follows.

Results (P.14, line 249 – P.15, line 254)

We then acquired the ^1H - ^{13}C HMQC spectra of MOR/ $\Delta 6\text{M}$ (DAMGO) and analyzed NMR signals from the methyl group of M283^{6,36}, which is located at the intracellular tip of TM6 (Fig. 4a) and exhibits different chemical shift values in the inactivated, partially-activated, and fully-activated conformations, serving as an ideal probe for the quantification of the populations of the three conformations (Fig. 4b and **Supplementary Fig. 7**).

Discussion (P.16, line 288 – P.17, line 296):

In this study, we analyzed the three-dimensional structure of MOR(DAMGO)- G_i -scFv16 in the BMS-986122 bound state and showed that BMS-986122 binds to TM3,

TM4, and TM5 from the lipid membrane side. The main chain traces of MOR were virtually identical between the cryo-EM structures in the absence and presence of BMS-986122. This similarity is likely because the G_i protein alters the overall mainchain conformation of MOR to the G_i protein-bound form, largely canceling the structural and dynamic heterogeneity in the intracellular effector-unbound states, which regulate the function of GPCRs^{1,19}, as observed by the NMR analysis of MOR in the mini-G_{s/i1} bound state (Supplementary Fig. 7).

Methods (P.30 line 528 – P.31 line 552):

Preparation of mini-G_{s/i1}

The protein sequence for mini-G_{s/i1} was derived from the preceding article⁵³. The DNA sequence encoding mini-G_{s/i1} with an N-terminal His tag followed by a TEV protease cleavage sequence was inserted into the pET-43a vector (Novagen). The *E. coli* strain BL21(DE3) was transformed with the resultant vector and cultured in LB medium containing 50 mg/L ampicillin. Protein expression was induced by the addition of isopropyl β-D-1-thiogalactopyranoside (IPTG) to a final concentration of 50 μM when the OD₆₀₀ reached 0.8. After 20 hours of culture at 25°C, cells were harvested by centrifugation at 9,000g for 15 minutes. The cells were resuspended in 100 mL of buffer containing 40 mM HEPES-NaOH (pH 7.5), 100 mM NaCl, 10% glycerol, 5 mM MgCl₂, 50 μM GDP, 28 μM E64, 10 μM leupeptin, 2.5 μM pepstatin A, 0.3 μM aprotinin, and 1 mM AEBSF. Cells were then disrupted by sonication and centrifuged at 38,000g for 45 minutes at 4°C. The supernatant was applied to His Select Nickel Affinity Gel (Sigma) and washed with a buffer containing 20 mM HEPES-NaOH (pH 7.5), 500 mM NaCl, 10% glycerol, 1 mM MgCl₂, 50 μM GDP, 1 mM DTT, and 40 mM imidazole. The His-

tagged mini-G_{s/i1} was then eluted from the column with a buffer containing 20 mM HEPES-NaOH (pH 7.5), 100 mM NaCl, 10 % glycerol, 1 mM MgCl₂, 50 μM GDP, 1 mM DTT, and 500 mM imidazole. The fractions containing His-tagged mini-G_{s/i1} were pooled and dialyzed overnight against a buffer containing 20 mM HEPES-NaOH (pH 7.5), 100 mM NaCl, 10 % glycerol, 1 mM MgCl₂, 50 μM GDP, and 1 mM DTT. EDTA was added to a final concentration of 0.5 mM, and the His-tag was cleaved by TEV protease for 16 hours at 4°C. The sample was then passed through His Select Nickel Affinity Gel, and the flow-through fraction was concentrated using Amicon Ultra-15 (MWCO 10 kDa). The concentrated sample was then purified by size exclusion chromatography using a HiLoad 26/600 Superdex 75 pg (Cytiva).

References

1. Shimada, I., Ueda, T., Kofuku, Y., Eddy, M. T. & Wüthrich, K. GPCR drug discovery: integrating solution NMR data with crystal and cryo-EM structures. *Nat Rev Drug Discov* **18**, 59–82 (2018).
19. Kaneko, S. *et al.* Activation mechanism of the μ-opioid receptor by an allosteric modulator. *Proc Natl Acad Sci U S A* **119**, e2121918119 (2022).
53. Nehmé, R. *et al.* Mini-G proteins: Novel tools for studying GPCRs in their active conformation. *PLoS One* **12**, e0175642 (2017).

Supplementary Fig. 7 | ^1H - ^{13}C HSQC spectra of MOR/ $\Delta 6\text{M}$ (DAMGO) in complex with mini- $\text{G}_{\text{s/i1}}$

a ^1H - ^{13}C HMQC spectra of MOR/ $\Delta 6\text{M}$ (DAMGO). Black: MOR/ $\Delta 6\text{M}$ in the full agonist DAMGO-bound state, red: MOR/ $\Delta 6\text{M}$ in the full agonist DAMGO-bound state in complex with the engineered G_i protein, mini- $\text{G}_{\text{s/i1}}$. In the mini- $\text{G}_{\text{s/i1}}$ bound state, only one signal was observed from $\text{M283}^{6,36}$, indicating that the structural equilibrium among the fully-activated (FA), partially-activated (PA), and the inactivated (I) conformations is suppressed by the binding of mini- $\text{G}_{\text{s/i1}}$.

b A closeup view of the cryo-EM structure of MOR(DAMGO)- G_i -scFv16 (PDB ID: 6DDF). The $\text{C}\epsilon$ atom of $\text{M283}^{6,36}$ is shown as a violet sphere, which is on the interface with the $\text{G}\alpha_i$ in the trimeric G_i protein, indicating that the chemical shift of the methyl group would be directly affected by the binding of the G_i protein or mini- $\text{G}_{\text{s/i1}}$.

Reviewer#1-2f) It's also a shame that the paper and the previous PNAS paper doesn't reference other NMR work that has actually dealt with multiple activation states in the presence and absence of G protein. For example, Huang et al, Cell, 2021 reveals three activation states, two of which are associated with partial and full agonism, and have fully qualified the meaning of active states and activation intermediates, with regard to G protein coupling.

We apologize for the oversight in not referencing previous NMR studies in the original manuscript. Following the reviewer's suggestion, we have now included references to other NMR studies that have explored multiple activation states in the presence and absence of G protein, notably the study by Huang *et al.*, *Cell* (2021).

Introduction (P.6, lines 95 – 98):

The observation of multiple states in the absence of G proteins, where some of them structurally resemble the G protein-bound states, has been observed for some other GPCRs²⁰⁻²⁵, indicating a common mechanism of activation shared in class A GPCRs.

References

20. Manglik, A. *et al.* Structural Insights into the Dynamic Process of β 2-Adrenergic Receptor Signaling. *Cell* **161**, 1101–1111 (2015).
21. Gregorio, G. G. *et al.* Single-molecule analysis of ligand efficacy in β 2AR-G-protein activation. *Nature* **547**, 68–73 (2017).
22. Kim, T. H. *et al.* The role of ligands on the equilibria between functional states of a G protein-coupled receptor. *J Am Chem Soc* **135**, 9465–74 (2013).
23. Solt, A. S. *et al.* Insight into partial agonism by observing multiple equilibria for ligand-bound and Gs-mimetic nanobody-bound β 1-adrenergic receptor. *Nat Commun* **8**, 1795 (2017).
24. Dixon, A. D. *et al.* Effect of Ligands and Transducers on the Neurotensin Receptor 1 Conformational Ensemble. *J Am Chem Soc* **144**, 10241–10250 (2022).
25. Huang, S. K. *et al.* Delineating the conformational landscape of the adenosine A2A receptor during G protein coupling. *Cell* **184**, 1884-1894.e14 (2021).

Reviewer#1-2g) The presence of two activation states, one of which is associated with lower efficacy, seems to be quite common and many believe there are general reasons for this.

We appreciate the reviewer's insightful observation on the potential general reasons underlying the phenomenon of GPCRs exhibiting two activation states with differing efficacies. This perspective has been integrated into the Discussion section, emphasizing its relevance in the rational design of allosteric modulators for other GPCRs, as follows:

Discussion (P.19, line 341 – P.20, line 345):

Considering the conservation of these residues (Supplementary Fig. 6), and the common observation of multiple states with varying activities across many GPCRs^{19–25}, the mechanism of allosteric activation of MOR by BMS-986122, as reported herein, may reflect a broader activation paradigm in other GPCRs.

References

19. Kaneko, S. *et al.* Activation mechanism of the μ -opioid receptor by an allosteric modulator. *Proc Natl Acad Sci U S A* **119**, e2121918119 (2022).
20. Manglik, A. *et al.* Structural Insights into the Dynamic Process of β 2-Adrenergic Receptor Signaling. *Cell* **161**, 1101–1111 (2015).
21. Gregorio, G. G. *et al.* Single-molecule analysis of ligand efficacy in β 2AR-G-protein activation. *Nature* **547**, 68–73 (2017).
22. Kim, T. H. *et al.* The role of ligands on the equilibria between functional states of a G protein-coupled receptor. *J Am Chem Soc* **135**, 9465–74 (2013).
23. Solt, A. S. *et al.* Insight into partial agonism by observing multiple equilibria for ligand-bound and Gs-mimetic nanobody-bound β 1-adrenergic receptor. *Nat Commun* **8**, 1795 (2017).
24. Dixon, A. D. *et al.* Effect of Ligands and Transducers on the Neurotensin Receptor 1 Conformational Ensemble. *J Am Chem Soc* **144**, 10241–10250 (2022).
25. Huang, S. K. *et al.* Delineating the conformational landscape of the adenosine A2A receptor during G protein coupling. *Cell* **184**, 1884–1894.e14 (2021).

Reviewer#1-3a) The authors write "However, the structural basis underlying the activity enhancement by the allosteric modulators is still largely unknown, hampering the rational design of allosteric modulators." This is firstly, a sweeping statement. One can find many papers that address allostery through MD, Monte Carlo analysis, and computational methods that hone in on allostery.

We appreciate the reviewer's comment that our initial statement may have been overly broad. We have revised the sentence to better represent the current state of research on allosteric modulation, acknowledging the valuable contributions from studies that utilize molecular dynamics (MD) simulation, Monte Carlo analysis, and computational methods.

Introduction (P.4, lines 62 – 66):

Although there are examples that have studied the mechanism of allosteric modulation of GPCRs using *in silico* approaches^{10–13}, the structural basis underlying the activity enhancement by the allosteric modulators remains incompletely understood. This gap in knowledge could potentially hinder the rational design of allosteric modulators.

References

10. Bartuzi, D., Kaczor, A. A. & Matosiuk, D. Activation and Allosteric Modulation of Human μ Opioid Receptor in Molecular Dynamics. *J Chem Inf Model* **55**, 2421–34 (2015).
11. Bartuzi, D., Kaczor, A. A. & Matosiuk, D. Interplay between Two Allosteric Sites and Their Influence on Agonist Binding in Human μ Opioid Receptor. *J Chem Inf Model* **56**, 563–70 (2016).
12. Hu, X., Provasi, D., Ramsey, S. & Filizola, M. Mechanism of μ -Opioid Receptor-Magnesium Interaction and Positive Allosteric Modulation. *Biophys J* **118**, 909–921 (2020).

13. Shpakov, A. O. Allosteric Regulation of G-Protein-Coupled Receptors: From Diversity of Molecular Mechanisms to Multiple Allosteric Sites and Their Ligands. *Int J Mol Sci* **24**, (2023).

Reviewer#1-3b) Secondly, if the goal is to identify R167-Y254 as a key activation switch associated with allostery, would it not make sense to invoke any of the above computational methods to understand its dynamic and hence, allosteric role?

Yes, it makes sense to employ these computational methods to understand the mechanism of the allosteric modulation of MOR by BMS-986122. Following the suggestion, we conducted an MD simulation of MOR in complex with the full agonist DAMGO and the allosteric modulator BMS-986122, using NAMD (Fig. R1). Given that the residence time of BMS-986122 on MOR is short in the MD timescale ($\sim 1 \mu\text{s}$), possibly reflecting its weak affinity ($EC_{50} \sim 10 \mu\text{M}$ (Kaneko *et al.*, *Proc Natl Acad Sci USA* (2022) **119**:e2121918119), and the time scale of exchange among the fully-activated, partially-activated, and inactivated conformations in the absence of G protein is estimated to be slower than 10 ms, it was not possible to recapitulate the experimentally observed phenomena in the MD simulation. Therefore, what we observed in the MD simulation was primarily the dissociation and reassociation of BMS-986122 during the simulation of $1 \mu\text{s}$ (Fig. R1). This exemplifies the importance of experimental observation of MOR to understand the mechanism of allosteric modulation by BMS-986122.

Fig. R1 | MD simulation trajectories of MOR(DAMGO)-Gi-scFv16 in complex with BMS-986122

The cryo-EM structure of MOR(DAMGO)-Gi-scFv16 in complex with BMS-986122 was subjected to molecular dynamics simulations using NAMD. Two trajectories of the distances between C06 atom of BMS-986122 and the C γ 2 atom of T162 are plotted against time in green (trajectory 1) and red (trajectory 2). In the bound state, the distance is around 5 Å. In trajectory 2, the dissociation-association of BMS-986122 on MOR was observed over time, whereas in trajectory 1, the dissociated BMS-986122 did not bind to MOR again (0.3-1.0 μs). This reflects the weak affinity (estimated to be 10 μM (Kaneko *et al.*, *Proc Natl Acad Sci USA* (2022) **119**:e2121918119), making it difficult to study the slow structural transition of MOR using molecular dynamics simulations.

We are grateful to the reviewer for raising this important point. We have incorporated these points into the manuscript as follows:

Discussion (P.18, lines 315 – 321):

It should be stressed here that, in our hands, molecular dynamics simulations were not able to recapitulate the structural transition of the three conformations, possibly because

the process occurs on a timescale slower than 10 ms (estimated from the signal splitting in Hz) which is much slower than the window of simulation time (1 μ s), thus making it difficult to study the allosteric modulation pathway by BMS-986122 *in silico*. This notion corroborates the importance of the experimental evidence of the dynamic properties of MOR by NMR.

Reviewer#1-4 " The mainchain traces of MOR were virtually identical between the cryo-EM structures in the absence and presence of BMS-986122. This similarity is probably because the Gi protein alters the overall mainchain conformation of MOR to the Gi protein bound form, canceling the structural and dynamic differences in the intracellular effector-unbound states, which regulate the function of GPCRs". I suggest a more cautious conclusion. The cryo-EM structure captures a single GDP-bound state. It's absolutely likely that dynamic equilibria are involved. Ideally, you'd need to extend the NMR observations to include mini-Gi to identify relative differences of functional states during G protein coupling, or capture cryo-EM states without nucleotide.

We appreciate the reviewer's suggestion for a more cautious conclusion regarding the comparison of cryo-EM structures of MOR in the absence and presence of BMS-986122. The reviewer correctly points out that the cryo-EM structure captures only a snapshot of the MOR structures, which are actually in the apo (nucleotide-free) form due to the removal of GDP through apyrase-mediated hydrolysis for stabilizing the MOR/G_i protein complex (Rasmusen *et al.*, *Nature* (2011) 477:549). We agree that dynamic equilibria likely exist in the G protein-bound state, recognizing that single-particle cryo-EM analysis may not capture the full spectrum of dynamics or structural heterogeneities.

Following the reviewer's advice, we extended our investigation to include NMR observations of MOR in the mini-G_{s/i1} bound state, which incorporates the G_i protein's C-terminal sequence within the mini-Gs scaffold known to interact with G_i-coupled GPCRs (Nehmé *et al.*, *PLoS ONE* (2017) 12:e0175642). As previously mentioned in the response to Reviewer#1-2e, M283 on TM6 displayed a single signal in the mini-G_{s/i1} bound state, contrary to three signals observed in the absence of mini-G_{s/i1} or G_i protein. This observation does not exclude the possibility of multiple conformations within the mini-Gi (or Gi protein) bound state but suggests a significant suppression of structural dynamics at the intracellular tips of TM6, which mediates the interaction with the Gi protein, due to the interaction with mini-G_{s/i1}.

In response to the reviewer's input, we revised the main text and included Supplementary Fig. 7. This amendment, prompted by the reviewer's insightful comments, clarifies that the cryo-EM single-particle analyses likely represent an averaged structure within the G_i protein-bound state, while the observed suppression in structural dynamics is substantiated by our additional NMR findings, corroborating the importance of investigating MOR in its dynamic and functional state in the absence of G proteins, using NMR.

Results (P.14, line 249 – P.15, line 254):

We then acquired the ¹H-¹³C HMQC spectra of MOR/Δ6M(DAMGO) and analyzed NMR signals from the methyl group of M283^{6,36}, which is located at the intracellular tip of TM6 (Fig. 4a) and exhibits different chemical shift values in the inactivated, partially-activated, and fully-activated conformations, serving as an ideal probe for the quantification of the populations of the three conformations (Fig. 4b and Supplementary Fig. 7).

Discussion (P.16 line 288 – P.17 line 296):

In this study, we analyzed the three-dimensional structure of MOR(DAMGO)-G_i-scFv16 in the BMS-986122 bound state and showed that BMS-986122 binds to TM3, TM4, and TM5 from the lipid membrane side. The main chain traces of MOR were virtually identical between the cryo-EM structures in the absence and presence of BMS-986122. This similarity is likely because the G_i protein alters the overall mainchain conformation of MOR to the G_i protein-bound form, largely canceling the structural and dynamic heterogeneity in the intracellular effector-unbound states, which regulate the function of GPCRs^{1,19}, as observed by the NMR analysis of MOR in the mini-G_{s/i1} bound state (Supplementary Fig. 7).

Methods (P.30 line 528 – P.27 line 552):

Preparation of mini-G_{s/i1}

The protein sequence for mini-G_{s/i1} was derived from the preceding article⁵³. The DNA sequence encoding mini-G_{s/i1} with an N-terminal His tag followed by a TEV protease cleavage sequence was inserted into the pET-43a vector (Novagen). The *E. coli* strain BL21(DE3) was transformed with the resultant vector and cultured in LB medium containing 50 mg/L ampicillin. Protein expression was induced by the addition of isopropyl β-D-1-thiogalactopyranoside (IPTG) to a final concentration of 50 μM when the OD₆₀₀ reached 0.8. After 20 hours of culture at 25°C, cells were harvested by centrifugation at 9,000g for 15 minutes. The cells were resuspended in 100 mL of buffer containing 40 mM HEPES-NaOH (pH 7.5), 100 mM NaCl, 10% glycerol, 5 mM MgCl₂, 50 μM GDP, 28 μM E64, 10 μM leupeptin, 2.5 μM pepstatin A, 0.3 μM aprotinin, and 1 mM AEBSF. Cells were then disrupted by sonication and centrifuged at 38,000g for 45

minutes at 4°C. The supernatant was applied to His Select Nickel Affinity Gel (Sigma) and washed with a buffer containing 20 mM HEPES-NaOH (pH 7.5), 500 mM NaCl, 10% glycerol, 1 mM MgCl₂, 50 μM GDP, 1 mM DTT, and 40 mM imidazole. The His-tagged mini-G_{s/i1} was then eluted from the column with a buffer containing 20 mM HEPES-NaOH (pH 7.5), 100 mM NaCl, 10 % glycerol, 1 mM MgCl₂, 50 μM GDP, 1 mM DTT, and 500 mM imidazole. The fractions containing His-tagged mini-G_{s/i1} were pooled and dialyzed overnight against a buffer containing 20 mM HEPES-NaOH (pH 7.5), 100 mM NaCl, 10 % glycerol, 1 mM MgCl₂, 50 μM GDP, and 1 mM DTT. EDTA was added to a final concentration of 0.5 mM, and the His-tag was cleaved by TEV protease for 16 hours at 4°C. The sample was then passed through His Select Nickel Affinity Gel, and the flow-through fraction was concentrated using Amicon Ultra-15 (MWCO 10 kDa). The concentrated sample was then purified by size exclusion chromatography using a HiLoad 26/600 Superdex 75 pg (Cytiva).

References

53. Nehmé, R. *et al.* Mini-G proteins: Novel tools for studying GPCRs in their active conformation. *PLoS One* **12**, e0175642 (2017).

Supplementary Fig. 7 | ^1H - ^{13}C HSQC spectra of MOR/ Δ 6M(DAMGO) in complex with mini- $\text{G}_{s/i1}$

a ^1H - ^{13}C HMQC spectra of MOR/ Δ 6M(DAMGO). Black: MOR/ Δ 6M in the full agonist DAMGO-bound state, red: MOR/ Δ 6M in the full agonist DAMGO-bound state in complex with the engineered G_i protein, mini- $\text{G}_{s/i1}$. In the mini- $\text{G}_{s/i1}$ bound state, only one signal was observed from M283^{6,36}, indicating that the structural equilibrium among the fully-activated (FA), partially-activated (PA), and the inactivated (I) conformations is suppressed by the binding of mini- $\text{G}_{s/i1}$.

b A closeup view of the cryo-EM structure of MOR(DAMGO)- G_i -scFv16 (PDB ID: 6DDF). The C ϵ atom of M283^{6,36} is shown as a violet sphere, which is on the interface with the $\text{G}\alpha_i$ in the trimeric G_i protein, indicating that the chemical shift of the methyl group would be directly affected by the binding of the G_i protein or mini- $\text{G}_{s/i1}$.

Reviewer#1-5) "Considering the conservation of these residues (Supplementary Fig. 5), the activation mechanism of MOR reported here might underlie the activation processes in many GPCRs." I suggest the authors evaluate active state complexes of various class A receptors with their cognate G proteins and show if there is a prevalent R167-Y254 hydrogen bond or not. Both residues are after all important in polar networks and the NPXXY motif.

We appreciate the reviewer's suggestion to evaluate the structures of various class A receptors with their cognate G proteins to determine the prevalence of the R^{3.50}-Y^{5.58} hydrogen bond. In response to the suggestion, we examined all class A GPCR structures in complex with G proteins or mini-G proteins listed in the GPCR database (<https://gpcrdb.org/>) as of December 23rd, 2023. Out of 408 structures, 339 feature the GPCR structure containing Arg at position 3.50 and Tyr at position 5.58, with side chain atoms modeled. To assess the potential for hydrogen bonds between these residues, we measured the inter-atomic distances of R^{3.50} C ζ (at the center of the guanidino group) and Y^{5.58} O atoms (Supplementary Fig. 10).

Supplementary Fig. 10 | Histogram of the distances between R^{3.50} and Y^{5.58} atoms in

GPCR structures

A histogram of the distances between R^{3.50} NH1/NH2/NE and Y^{5.58} O atoms in class A GPCRs in complex with G proteins or mini-G proteins. For R^{3.50}, the closest nitrogen atom to the Y^{5.58} O atom is selected for each structure. As of December 23rd, 2023, the GPCR database (<https://gpcrdb.org/>) lists 408 structures of GPCRs in complex with G proteins or mini-G proteins. Among them, 339 are of GPCRs harboring Arg at 3.50 and Tyr at 5.58. For these 339 structures, distances between the nitrogen atoms of R^{3.50} in the side chain guanidino group, which can be the proton acceptor, and the oxygen atom of Y^{5.58}, which can be the proton donor, were calculated, and the smallest distance is used. Respective structures are shown as insets, Structures with possible hydrogen bonding between these two residues are highlighted in red boxes.

Given the moderate resolutions of the structures (2.0-4.5Å), unambiguously identifying hydrogen bonds based solely on distance can be challenging. However, we applied a threshold of 3.0 Å, representing the mean distance between nitrogen and oxygen atoms in N...H-O type hydrogen bonds (de Freitas and Schapira *MedChemComm* (2017) 8:1970). Among the 339 structures, 106 exhibited inter-atomic distances less than 3.0 Å, indicating that approximately 25% of the currently available GPCR structures in complex with G proteins or mini-G proteins may feature possible hydrogen bonds between R^{3.50} and Y^{5.58}.

In the cryo-EM structures reported in our study, the distances between the nitrogen of R^{3.50} and the oxygen of Y^{5.58} are 3.3 Å and 2.6 Å in the absence and presence of BMS-986122, respectively. This suggests a higher likelihood of hydrogen bond formation between these residues in the presence of BMS-986122, prompting our focus on these interactions. Our study concludes that the formation of this hydrogen bond plays an important role in stabilizing the fully-activated state in the absence of G proteins —prior to interaction with the Gi protein— not necessarily when in complex with G proteins. Since these residues are at the interface with G proteins, the binding of G proteins could interfere with their interaction, making it impractical to deduce the significance of this

interaction from complex structures solved by cryo-EM or X-ray crystallography.

We value the reviewer's input and have added the following sentence to the Discussion section to reflect this. This amendment allows us to propose that the interaction might be observable in previous structures, although our findings highlight its importance in the dynamic structural equilibrium that governs the apparent activity of GPCRs in the absence of G proteins or β -arrestins.

Discussion (P.17, lines 301 – 306):

A survey in the Protein Data Bank (PDB) database for class A GPCR structures in complex with their cognate G proteins or mini-G proteins revealed that in 106 out of 339 structures, distances between N in R^{3.50} and O in Y^{5.58} are closer than 3.0 Å, suggesting that the hydrogen bonding interaction between these residues is not specific to MOR⁴² (Supplementary Fig. 10).

References

42. Ferreira de Freitas, R. & Schapira, M. A systematic analysis of atomic protein-ligand interactions in the PDB. *Medchemcomm* **8**, 1970–1981 (2017).

Supplementary Fig. 10 | Histogram of the distances between R^{3.50} and Y^{5.58} atoms in GPCR structures

A histogram of the distances between R^{3.50} NH1/NH2/NE and Y^{5.58} O atoms in class A GPCRs in complex with G proteins or mini-G proteins. For R^{3.50}, the closest nitrogen atom to the Y^{5.58} O atom is selected for each structure. As of December 23rd, 2023, the GPCR database (<https://gpcrdb.org/>) lists 408 structures of GPCRs in complex with G proteins or mini-G proteins. Among them, 339 are of GPCRs harboring Arg at 3.50 and Tyr at 5.58. For these 339 structures, distances between the nitrogen atoms of R^{3.50} in the side chain guanidino group, which can be the proton acceptor, and the oxygen atom of Y^{5.58}, which can be the proton donor, were calculated, and the smallest distance is used. Respective structures are shown as insets, Structures with possible hydrogen bonding between these two residues are highlighted in red boxes.

Reviewer#1-6a) Presumably pharmacologists would ask why full efficacy hasn't evolved for the endogenous opioid receptor. Could it be that the BMS-activated state leads to greater desensitization over time?

Yes, it is conceivable that a BMS-activated-like state, if it existed endogenously, could lead to greater desensitization over time. BMS-986122 shifts the intrinsic structural equilibrium between the inactive, partially-activated, and fully-activated states, regulating the overall activity of MOR in the absence of intracellular binders. This shift can affect not only the G protein signaling but also other signaling pathways, such as those involving β -arrestin, which could lead to MOR desensitization. However, because the signaling level should be precisely balanced by various factors, such as the concentration of endogenous agonists or the number of receptors, it is also conceivable that the level of efficacy does not necessarily need to be higher than that of MOR without allosteric modulators. Therefore, there may not have been any evolutionary pressure for nature to select for full efficacy of MOR.

Reviewer#1-6b) Is the BMS-activated state more prone to phosphorylation?

Yes, it is conceivable that the BMS-activated-like state may be more susceptible to phosphorylation, for reasons similar to those outlined in the response to Reviewer#1-6a. The alteration in the equilibrium could lead to changes in GRK-mediated phosphorylation as well. It is also possible that, from an evolutionary perspective, there has not been a necessity for higher efficacy beyond the current level in the absence of an allosteric modulator.

Reviewer#1-6c) Or might it be that *in vivo*, there are other adjuvants that rescue high efficacy signalling?

Yes, it is conceivable that *in vivo*, there may be other adjuvants that enhance the efficacy of MOR. One possibility includes lipid molecules that interact with the transmembrane region where BMS-986122 binds, thereby modulating the apparent efficacy of MOR. This notion is supported by evidence that the activity of GPCRs can be allosterically modulated by lipid molecules (van der Westhuizen *et al.*, *J Pharmacol Exp Ther* (2015) 353:246).

Reviewer#1-6d) The authors could nicely round out this paper if they were to add a paragraph or two to the discussion, bringing in the pharmacology of BMS-986122 in light of their observations.

We appreciate the reviewer's insightful suggestion and have accordingly added the following paragraphs to the Discussion section. This enhancement, we believe, broadens the appeal of the manuscript to a wider readership, including pharmacologists.

Discussion (P.21, lines 362 – 379):

In cells, the signaling of GPCRs is complexly controlled based on various factors including the expression levels of these receptors and the concentrations of endogenous agonists. This complexity makes it challenging to understand if there are evolutionary reasons why some GPCRs have evolved so that they do not elicit maximal activity

solely via the binding of orthosteric ligands. However, it is noteworthy from a pharmaceutical standpoint that the concomitant use of allosteric modulators with orthosteric ligands can elicit higher activity in some GPCRs than would be expected from the efficacy of endogenous ligands alone. This is reflected in the NMR spectra of MOR in the full agonist-bound state, where the population of the fully-activated conformation is not 100% (Fig. 4b). It may be possible to evaluate if allosteric modulation would be feasible for a certain GPCR for which allosteric modulators have not been identified, by observing the NMR spectra thereof in the full agonist-bound state. Identifying and controlling the function-related structural equilibria by analyzing NMR spectra, while complementing the structural information of the compound's binding site by cryo-EM analyses, presents an attractive strategy for the development of novel GPCR therapeutics, enabling the full activation of such GPCRs. We envision that the findings reported here may accelerate the development of novel analgesics targeting MOR, and therapeutics for diseases related to GPCRs.

Reviewer #2:

In this study, the authors present the cryo-EM structure of a DAMGO (full agonist) – MOR (μ -opioid receptor) – Gi – BMS-986122 (allosteric modulator) complex. They utilized NMR data to ascertain the orientation of BMS-986122 within this structure. Comparing the MOR–Gi structures with and without BMS-986122, a connecting density between residues R167(3.50) and Y254(5.58) was observed only in the BMS-986122-bound state. Further, NMR was employed to track conformational shifts in M257, near Y254, and M283 at the cytoplasmic end of TM6, revealing distinct MOR conformations modulated by BMS-986122.

While the exploration of the molecular mechanism of allosteric modulators in GPCRs is compelling, the conclusions drawn are not sufficiently supported by the data presented.

We would like to express our gratitude to the reviewer for the thorough review of our manuscript.

We appreciate the constructive comments and have carefully revised the manuscript in response.

Below, we address each of the reviewer's comments and suggestions as follows:

Major Comments:

#2-major1. The authors suggest a transient interaction between R167(3.50) and Y254(5.58) is enhanced by BMS-986122, contributing to MOR activation. However, this interaction is one among many in the active state of class A GPCRs. Without examining all interactions stabilizing MOR's active state, attributing BMS-986122's effect solely to this interaction is speculative. A more cautious conclusion might be that BMS-986122 stabilizes the active state conformation, akin to other positive allosteric modulators (e.g., PMID: 34497422).

We appreciate the reviewer's observation that our initial conclusion may have appeared speculative. As the reviewer pointed out, the interaction between R167^{3.50} and Y254^{5.58} is one among many interactions in the active state of class A GPCRs. We also agree with the reviewer's comment that BMS-986122 may stabilize the active state conformation as a whole, similar to other positive allosteric modulators. Our conclusion in this article aligns with these statements. From the NMR analyses and activity assay (Fig. 4 b and c, shown below for your reference), we propose that the interaction between R167^{3.50} and Y254^{5.58} is involved in the stabilization of the fully-activated state in the three-state equilibrium occurring in the absence of G proteins. Here the term 'stabilization' is used to indicate that the hydroxy group of Y254^{5.58} makes the fully-activated conformation more populated in the equilibrium by decreasing the free energy of the state, and not that it is the sole keystone for MOR to maintain the active conformation in the cryo-EM structure of MOR in complex with G_i protein and BMS-986122. Indeed, the Y254^{5.58}F mutation abolished the fully-activated conformation and abrogated the activity of MOR, indicating that the hydroxy group of Y254^{5.58} plays a pivotal role in MOR activation. However, the addition of BMS-986122 to the Y254^{5.58}F variant partly rescued the fully-activated conformation and the activity of MOR, corroborating that the hydroxy group is not imperative for the activation of MOR and that the Y254^{5.58}F mutation does not abolish the interaction with the G protein. Since a single mutation that removes the hydroxy group from Y254^{5.58} completely destabilizes the fully-activated conformation (Fig. 4b), without deteriorating the whole structure or interaction with the G protein (Fig. 4c), we conclude that the interaction involving the hydroxy group of Y254 is one of the most important, if not imperative, to stabilize the fully-activated conformation of MOR in which TM6 is outward-shifted in the absence of G proteins.

Fig. 4 | Importance of R167^{3.50}-Y254^{5.58} hydrogen bonding interaction for function and structural dynamics of MOR

a Intracellular view of structures of MOR bound to DAMGO and BMS-986122 (left) and bound to antagonist β -FNA (right, PDB ID: 4DKL). TM6 helices of MOR were colored green and magenta, and the C ϵ atoms of M283 were represented by spheres.

b ¹H-¹³C HMQC spectra of the parental MOR/ Δ 6M (left) or the Y254^{5.58}F variant (right) bound to DAMGO in the presence and absence of BMS-986122.

c GTP turnover of the G_i protein stimulated by MOR/ Δ 6M or the Y254^{5.58}F variant in the DAMGO-bound state in the presence and absence of BMS-986122. All values were normalized by the GTP turnover rate of the G_i protein in the presence of MOR/ Δ 6M in the full agonist DAMGO-bound state. Data are presented as mean \pm standard error of

the mean (s.e.m.) (n=3 independent replicates). Statistical significance was determined by a one-tailed Student's t-test. *: $p < 0.05$.

In our model, the structural role of the hydroxy group of Y254^{5,58} is to increase the population of the fully-activated conformation in the structural equilibrium, i.e., the conformation where TM6 is outward-shifted, by preventing the closing of TM6, thus increasing the apparent on-rate of the interaction with the G_i protein (Fig. 5). Although the binding site is different, this may be conceptually akin to the MD simulations in the preceding article on another class A GPCR, A1R, and its allosteric modulator MIPS521 (Fig. 4 e and f in Draper-Joyce *et al.*, *Nature* (2021) 597:571 (PMID: 34497422), shown below for your reference). We appreciate the reviewer for raising this point and apologize for any confusion our original manuscript may have caused. In accordance with the reviewer's suggestion, we have revised the manuscript as follows, referring to the previous articles including Draper-Joyce *et al.*, *Nature* (2021). Thanks to the constructive suggestion, we hope that the current manuscript is comprehensive and acceptable to the readership of *Nature Communications*.

Fig. 5 | The role of R167^{3.50}-Y254^{5.58} hydrogen bonding interaction in the function-related equilibrium of MOR

A schematic model illustrating the effects of the hydrogen bonding interaction between R167^{3.50} and Y254^{5.58} on the conformational equilibrium of MOR is shown.

[REDACTED]

Fig. 4 in Draper-Joyce et al., *Nature* (2021) 597:571 | MIPS521 stabilizes the A1R–Gi2 ternary complex. e and f, Distance between the intracellular ends of TM3 and TM6

(measured as the distance in Å between R105^{3.50} and E229^{6.30}) in the absence (e) or presence (f) of MIPS521. Each condition represents three GaMD simulations, with each simulation trace displayed in a different colour (black, red, blue). The lines depict the running average over 2 ns.

Discussion (P.17, line 306 – P.18, line 315):

Although this should not be the only structural difference caused by the binding of BMS-986122, our solution NMR analyses of MOR in the absence of G_i proteins demonstrated that the conformational equilibrium around Y254^{5.58} is suppressed by BMS-986122 and enhanced by the sole Y254^{5.58}F mutation that removes the hydroxy group from Y254^{5.58}. Concurrently, our NMR analyses revealed that the disruption of the hydrogen bonding interaction between R167^{3.50} and Y254^{5.58} decreases the population of the fully-activated conformation (Fig. 4b), where the intracellular half of TM6 in MOR is outward-shifted¹⁹. These suggest that the hydrogen bonding interaction between R167^{3.50} and Y254^{5.58} plays an important role in stabilizing the outward-shifted TM6, likely by blocking the inward shift of TM6 (Fig. 5).

Discussion (P.20, lines 346 – 357):

Previous structural analyses have showcased many allosteric modulator binding sites on various GPCRs¹⁴. Some of these studies have proposed a mechanism where the positive allosteric modulators stabilize the active state conformation (i.e., the conformation in the G protein-bound state) of GPCRs^{43–46}, but the structural basis of the stabilization remained uncharacterized, or different in each case. These imply that, compared to orthosteric ligands, there can be many structural pathways within GPCR molecules to allosterically modulate their activity. This observation makes it challenging to rationally

design a novel allosteric modulator for a certain GPCR. The mechanism of the allosteric modulation of MOR by BMS-986122 shown here illustrates that the binding of the allosteric modulator laterally rearranges the interhelical interaction between the transmembrane helices, thus altering the key interaction between the conserved residues, R167^{3.50} and Y254^{5.58}, at the core region.

References

43. Liu, X. *et al.* Mechanism of β 2AR regulation by an intracellular positive allosteric modulator. *Science* **364**, 1283–1287 (2019).
44. Zhuang, Y. *et al.* Mechanism of dopamine binding and allosteric modulation of the human D1 dopamine receptor. *Cell Res* **31**, 593–596 (2021).
45. Lu, J. *et al.* Structural basis for the cooperative allosteric activation of the free fatty acid receptor GPR40. *Nat Struct Mol Biol* **24**, 570–577 (2017).
46. Draper-Joyce, C. J. *et al.* Positive allosteric mechanisms of adenosine A1 receptor-mediated analgesia. *Nature* **597**, 571–576 (2021).

#2-major2. The GTP turnover assay indicates that the Y254F mutant retains approximately 48% activity with BMS-986122, suggesting the R167-Y254 interaction is not solely responsible for BMS-986122's efficacy.

Yes, as the reviewer pointed out, the Y254^{5.58}F mutant retains approximately 48% activity with BMS-986122 (Fig. 4c, shown below for your reference). This suggests that the R167^{3.50}-Y254^{5.58} interaction is not solely responsible for BMS-986122's efficacy, as previously discussed in response to Reviewer#2-major1. We interpret the data as evidence that the hydroxy group of Y254 plays a crucial role in the activation of MOR, as the mutation reduced activity both in the absence and presence of BMS-986122. The rescue of activity upon addition of BMS-986122 is significant

because it corroborates that the mutation did not compromise the structure of MOR nor its interaction with the G protein. In our proposed model, BMS-986122 enhances the role of the interaction in the wild type, whereas it partly compensates for the interaction in the Y254^{5.58}F variant through a direct interaction with the TM3 helix, where R167^{3.50} is located (Fig. 5, shown below for your reference).

Fig. 4 | Importance of R167^{3.50}-Y254^{5.58} hydrogen bonding interaction for function and structural dynamics of MOR

a Intracellular view of structures of MOR bound to DAMGO and BMS-986122 (left) and bound to antagonist β -FNA (right, PDB ID: 4DKL). TM6 helices of MOR were colored green and magenta, and the C ϵ atoms of M283 were represented by spheres.

b ¹H-¹³C HMQC spectra of the parental MOR/ Δ 6M (left) or the Y254^{5.58}F variant (right) bound to DAMGO in the presence and absence of BMS-986122.

c GTP turnover of the G_i protein stimulated by MOR/ Δ 6M or the Y254^{5.58}F variant in the DAMGO-bound state in the presence and absence of BMS-986122. All values were normalized by the GTP turnover rate of the G_i protein in the presence of MOR/ Δ 6M in the full agonist DAMGO-bound state. Data are presented as mean \pm standard error of the mean (s.e.m.) (n=3 independent replicates). Statistical significance was determined by a one-tailed Student's t-test. *: p < 0.05.

Fig. 5 | The role of R167^{3.50}-Y254^{5.58} hydrogen bonding interaction in the function-related equilibrium of MOR

A schematic model illustrating the effects of the hydrogen bonding interaction between

R167^{3.50} and Y254^{5.58} on the conformational equilibrium of MOR is shown.

#2-major3. Additional cryo-EM data is needed to increase the resolution and confirm BMS-986122's conformation in the structure.

We appreciate the reviewer's suggestion to increase the resolution to confirm BMS-986122's conformation in the structure. In accordance with the reviewer's suggestion, we have collected additional cryo-EM data of MOR(DAMGO)-G_i-scFv16 in the presence of BMS-986122 and performed another single-particle analysis for the new dataset (Figs. R2, R3, and Table R1). As a result, with more than twice as many particles (702,716 instead of 308,249), the resolution remained identical at 3.1Å, corroborating that the previous datasets had been sufficient for this system. For your reference, we submit the map with an increased number of particles, together with the revised manuscript.

Fig. R2 | Cryo-EM data acquisition and reprocessing of the MOR(DAMGO)-Gi-scFv16 complex with BMS-986122.

a Flowchart for cryo-EM data processing of the complex structure.

b Representative raw micrograph.

c Gallery of two-dimensional class averages.

d Final three-dimensional density map colored by local resolution in side views.

e FSC curves after post-processing in RELION.

Fig. R3 | Comparison of density maps

Comparison of the original density map from the single-particle analysis of 308,249 particles (left) with the newly obtained density map obtained from the additional data of 702,716 particles in total (right). Model resolutions are 3.05 and 3.09 Å, respectively. Densities corresponding to BMS-986122 and DAMGO are colored red and orange, respectively.

Table R1 | Cryo-EM data collection, refinement, and validation statistics

	MOR(DAMGO)- Gi-scFv16 complex	MOR(DAMGO)-Gi-scFv16 containing BMS-986122	
EMDB ID	EMD-36989	EMD-36990	
PDB ID	8K9K	8K9L	
Data collection and processing		Data-1	Data-2
Microscope	Titan Krios G4	Titan Krios G4	Titan Krios G4
Magnification	105,000	105,000	105,000
Voltage (kV)	300	300	300
Detector	Gatan	Gatan	Gatan
	BioQuantum K3	BioQuantum K3	BioQuantum K3
Movies	6,450	5,846	14,109
Electron dose (e-/Å ²)	50.5	50.5	50.5
Defocus range (µm)	-0.8 to -2.0	-0.8 to -2.0	-0.8 to -2.0
Effective pixel size (Å)	0.83	0.83	0.83
Initial number of particles	8,500,757	3,452,964	4,066,172
Final number of particles	802,844	308,249	394,467
			Merged *
Symmetry imposed	C1		C1
Map resolution (Å)	2.98		3.09
FSC threshold	0.143		0.143
Map resolution range (Å)	2.8-3.8		3.0-3.8
Refinement			
Initial model used (PDB code)	7SBF, 6CRK	8K9K	
Model resolution (Å)	2.98	3.09	
FSC threshold	0.143	0.143	
Map resolution range (Å)	2.8-3.8	3.0-3.8	
Map sharpening B -factor (Å ²)	-128.43	-116.82	
Nonhydrogen atoms	8,923	8,919	
Protein residues	1,142	1,142	
Mean B -factors (Å ²)			
Protein	57.04	82.68	
Ligand	60.87	134.09	
R.m.s. deviations			
Bond lengths (Å)	0.003	0.003	
Bond angles (°)	0.624	0.587	
Validation			
MolProbity score	1.63	1.55	
Clash score	7.57	6.56	
Rotamer outliers (%)	0.10	0.31	
Ramachandran plot			
Favored (%)	96.62	96.89	
Allowed (%)	3.38	3.11	
Outliers (%)	0.00	0.00	

* Statistics of final map after merging datasets 1 and 2.

We understand the reviewer's concern that the resolution is moderate to unambiguously model BMS-986122 solely by the density map. Therefore, when docking BMS-986122, we conducted NMR experiments to determine the relative orientation of BMS-986122 (Supplementary Fig. 4) and evaluated the involvement of W158^{3.41} and F180^{ICL2} in the interaction with BMS-986122 (Supplementary Fig. 5). We also performed an additional ligand binding assay in which W158^{3.41} and F180^{ICL2} are mutated, in response to the comment below (Reviewer#2-minor3) and confirmed that the F180^{ICL2}L mutation abrogated the sensitivity of MOR for the allosteric modulation by BMS-986122 (Supplementary Fig. 5c). All these data support the binding pose of BMS-986122 as reported herein.

Supplementary Fig. 4 | Binding site and interaction of BMS-986122

a Two possible orientations of BMS-986122 to MOR. The C ϵ atom of M245^{5.49} is represented by a sphere. The methoxy (blue) and bromo (red) groups are indicated by squares.

b The chemical structures of BMS-986122 (left) and BMS-986124 (right). The methoxy

(blue) and bromo (red) groups are indicated by squares.

c ^1H - ^{13}C HMQC signals of M245^A of MOR/ Δ 6M in DAMGO-bound (left), DAMGO and BMS-986122 bound (middle), and DAMGO and BMS-986124 bound (right) states. The M245^A signal was not observed for DAMGO and BMS-986122 bound MOR/ Δ 6M. **d** A closeup view of BMS-986122 in the cryo-EM structure of MOR(DAMGO)-G_i-scFv16 docked using the information obtained from the NMR analyses (**b** and **c**). MOR structure model is shown as a transparent green surface with stick and cartoon models.

Supplementary Fig. 5 | Aromatic residues of MOR involved in interaction with BMS-986122

a Comparison of the binding site of BMS-986122 in the presence (green) and absence (blue) of BMS-986122.

b ^1H - ^{13}C HMQC signals of M245^A and overlays of the cross section of the M245^A signal in the absence (black) and presence (green) of 100 μM BMS-986122. The M245^A signal was not observed for DAMGO and BMS-986122 bound MOR/ Δ 6M.

c Relative activities of MOR/ Δ 6M and its variants quantified by the GTP turnover assay. DAMGO: Full agonist DAMGO-bound state in the absence of BMS-986122, DAMGO+BMS-986122: DAMGO-bound state in the presence of BMS-986122. Data are presented as mean \pm standard error of the mean (s.e.m.). Statistical significance was determined by a one-tailed Student's t-test. *: $p < 0.05$, n.s.: not significant ($p > 0.05$). $n=3$ independent replicates for MOR/ Δ 6M and MOR/ Δ 6M/F180^{ICL2L}, and $n=2$ independent replicates for MOR/ Δ 6M/W158^{3.41M}.

In accordance with the reviewer's comment, we revised the manuscript as follows. Thanks to the amendments, we believe that how BMS-986122 is modeled in the cryo-EM structure is now more comprehensively provided to the readership.

Results (P. 9, lines 152 – 160):

However, due to the relatively weak EM density at this allosteric site, **as well as the moderate resolution of 3.1 Å, it was not possible to unambiguously model BMS-986122 solely by the cryo-EM data.** To determine the orientation of BMS-986122 relative to MOR, we compared the NMR spectra of MOR(DAMGO) bound to BMS-986122 and its analog BMS-986124 (Supplementary Fig. 4). BMS-986124 is an analog of BMS-986122, in which the positions of the bromo- and methoxy- moieties on the benzyl group are different, that competes with BMS-986122 on MOR but without allosteric modulation activity (Supplementary Fig. 4b)¹⁶.

Results (P. 10, lines 168-169):

We then ~~modeled~~ docked BMS-986122 in the cryo-EM structure so that the methoxybenzyl bromide moiety is adjacent to M245^{5,49} (Supplementary Fig. 4d).

Minor Comments:

#2-minor1. Page 3: Replace “extracellular active site” with “ligand binding site” for accuracy.

Thank you for your comment. We have replaced “extracellular active site” with “ligand binding site”, as suggested. We apologize for the inaccurate terminology used in the original manuscript.

Introduction (P.3, line 49 – 52)

Furthermore, previous functional studies on regulatory mechanisms of GPCRs demonstrated that the activity induced by orthosteric ligands—compounds that bind to ~~the extracellular active site~~ ligand binding site of a receptor—does not necessarily represent the full extent of GPCR function.

#2-minor2. Figure 1: Include the density for DAMGO, as was done for BMS-986122, for comparative purposes.

Thank you for the comment. We have included the density for DAMGO as was done for BMS-986122 in Figure 1, as below. Thanks to the change, the quality and fit of the density corresponding to DAMGO and BMS-986122 are readily comparable.

Fig. 1 | Overall structure of MOR in complex with a full agonist and an allosteric modulator

a A cryo-EM density map of the MOR(DAMGO)-G_i-scFv16 complex in the presence of BMS-986122. **Insets show expanded views of the DAMGO and BMS-986122 models and their corresponding density maps.** Densities are contoured at a threshold level of 2.4 in ChimeraX.

b The overall structure of the MOR(DAMGO)-G_i-scFv16 complex in the presence of BMS-986122.

#2-minor3. The manuscript would benefit from ligand binding data to substantiate claims regarding F180 and W158's interaction with BMS-986122.

We would like to thank the reviewer for the suggestion to add ligand binding data to substantiate claims regarding F180 and W158's interactions with BMS-986122. Following the suggestion, we conducted mutational analyses in which F180^{ICL2} and W158^{3,41} were substituted with Leu and Met, respectively. The F180^{ICL2}L variant lost the BMS-986122-dependent increase in activity, whereas the W158^{3,41}M variant retained susceptibility similar to the parental construct, MOR/Δ6M (Fig. R4). These results strongly support that F180 is involved in the interaction with BMS-986122. Although

the W158^{3,41}M variant did not show a discernible difference from the parental MOR/ Δ 6M construct in this assay, our NMR analyses clearly demonstrated that the W158^{3,41}M variant exhibits weakened affinity for BMS-986122, as evidenced by the observation of reduced NMR signal from M245 (Supplementary Fig. 5b). This is likely because the weakening of the affinity for BMS-986122 by the W158^{3,41}M mutation is small to be observed by the GTP turnover assay experiment, while the effect is clearly observed in NMR experiments that sensitively detect the weak interactions between molecules. Together, these results substantiate our claim regarding the interaction of these residues with BMS-986122.

Fig. R4 | GTP turnover assay of the mutants

Relative activity of MOR/ Δ 6M or its variants quantified by the GTP turnover assay. DAMGO: Full agonist DAMGO-bound state in the absence of BMS-986122, DAMGO+BMS-986122: DAMGO-bound state in the presence of BMS-986122. Data are presented as mean \pm standard error of the mean (s.e.m.).

Statistical significance was determined by one-sided Student's t test. *: $p < 0.05$, n.s.: not significant ($p > 0.05$). $n=3$ independent replicates for MOR/ $\Delta 6M$ and MOR/ $\Delta 6M$ /F180^{ICL2}L, and $n=2$ independent replicates for MOR/ $\Delta 6M$ /W158^{3,41}M.

We have added these results as Supplementary Fig. 5c and referred to them in the manuscript as follows. Thanks to the suggestion, the binding site and interaction of BMS-986122 with MOR and the involvement of F180^{ICL2} in the allosteric modulation by BMS-986122 are more strongly supported.

Results (P.10, line 171 – P.11, line 188):

To test whether these aromatic residues are involved in the interaction with BMS-986122, we analyzed the perturbation of the NMR signal of M245^{5,49} for W158^{3,41}M or F180^{ICL2}L variants upon the addition of BMS-986122, and the allosteric modulation effects of these variants (Supplementary Fig. 5, b and c). As mentioned above, the M245^A signal is broadened upon the addition of BMS-986122 in the parental MOR/ $\Delta 6M$ (DAMGO), while in the F180^{ICL2}L or W158^{3,41}M variants, the M245^A signal did not disappear in the presence of 100 μ M BMS-986122, indicating that these variants possess reduced affinities for BMS-986122 compared to the parental MOR/ $\Delta 6M$ (Supplementary Fig. 5b). These results indicate that the side chains of F180 and W158 are involved in the interaction with BMS-986122, which supports the cryo-EM structure of the MOR(DAMGO)-G_i-scFv16 complex in the BMS-986122 bound state. This is further confirmed by a GTP turnover assay that measures the consumption of GTP by G proteins through GTP to GDP turnover³⁴⁻³⁶ (Supplementary Fig. 5c). F180^{ICL2}L exhibited a loss of BMS-986122-dependent increase in the activity of MOR, while

W158^{3.41}M exhibited an increase in the activity upon the addition of BMS-986122 as the parental MOR/ Δ 6M, suggesting that F180^{ICL2} is more important for the allosteric modulation activity of BMS-986122 than aromatic residues at 3.41.

Supplementary Fig. 5 | Aromatic residues of MOR involved in interaction with BMS-986122

a Comparison of the binding site of BMS-986122 in the presence (green) and absence (blue) of BMS-986122.

b ¹H-¹³C HMQC signals of M245^A and overlays of the cross section of the M245^A

signal in the absence (black) and presence (green) of 100 μ M BMS-986122. The M245^A signal was not observed for DAMGO and BMS-986122 bound MOR/ Δ 6M.

c Relative activities of MOR/ Δ 6M and its variants quantified by the GTP turnover assay. DAMGO: Full agonist DAMGO-bound state in the absence of BMS-986122, DAMGO+BMS-986122: DAMGO-bound state in the presence of BMS-986122. Data are presented as mean \pm standard error of the mean (s.e.m.). Statistical significance was determined by a one-tailed Student's t-test. *: $p < 0.05$, n.s.: not significant ($p > 0.05$). $n=3$ independent replicates for MOR/ Δ 6M and MOR/ Δ 6M/F180^{ICL2L}, and $n=2$ independent replicates for MOR/ Δ 6M/W158^{3.41M}.

#2-minor4. Figure 2B: Given the ~ 3 Å resolution, the significance of differences in density maps is unclear, especially as side chains appear similarly oriented.

Yes, it is correct that with the resolution obtained here, the differences in density maps may not be significant. Even after the addition of cryo-EM data for better resolution (Reviewer#2-major3), the resolution of MOR(DAMGO)-Gi-ScFv16 in complex with BMS-986122 remained 3.1 Å, making it difficult to claim that the small difference in the structure is significant, especially since the side chains are similarly oriented, as the reviewer pointed out. However, the differences are notable in the density maps, indicating the difference in the interaction between R167^{3.50} and Y254^{5.58} (Fig. 2b). Inspired by the difference, we have shown that these residues play important roles in the dynamic structural equilibrium in the absence of G_i proteins predetermining the apparent activity of MOR, which cannot be investigated by cryo-EM or X-ray crystallography.

Fig. 2 | Comparison of structures of MOR(DAMGO) in the presence and absence of BMS-986122

a Comparison of cryo-EM structures of the MOR(DAMGO)-G_i-scFv16 complex in the presence (green) and absence (cyan) of BMS-986122.

b Overlay of cryo-EM density maps and structures around R167^{3.50}, Y254^{5.58}, and Y338^{7.53} residues of MOR(DAMGO) in the presence (left) and absence (right) of BMS-986122. The differences in density between R167^{3.50} and Y254^{5.58} are highlighted with red circles. Densities are contoured at 3.00 and 0.012 in ChimeraX for the maps in the presence and absence of BMS-986122, respectively, representing a significant difference in the contact region between R167^{3.50} and Y254^{5.58}.

In response to the reviewer's insightful comment, we revised the main text to describe the differences in density, not the conformations of the side chains or the distance between them, as follows.

Introduction (P. 7, lines 110 – 113):

Comparison of **the two density maps** in the presence and absence of BMS-986122 identifies a small difference induced by BMS-986122 binding, between the sidechains of highly conserved residues R167^{3,50} in the DRY motif and Y254^{5,58} involved in the water-mediated interaction with **the NPxxY motif**.

Results (P. 11, lines 189 – 194):

By comparing structural models of MOR(DAMGO) with and without BMS-986122, it has been shown that the positions of the main chain atoms are nearly identical (Fig. 2a, r.m.s.d. = 0.36 Å for all C α atoms of MOR). However, notable differences between **the density maps** in the absence and presence of BMS-986122 were observed **between the side chains of MOR residues in the structural motifs** that have been shown to play important roles in the activation of GPCRs (Fig. 2b).

Discussion (P. 16, lines 296 – 301):

Although cryo-EM analyses of agonist-activated GPCRs in the absence of intracellular binders are generally difficult due to smaller molecular sizes and inherent function-related dynamics, **comparison of the cryo-EM maps of MOR(DAMGO)-G_i-scFv16 in the absence and presence of BMS-986122 identified a notable difference** that may have remained after binding with the G_i protein, as in the **increased density** between the conserved residues R167^{3,50} and Y254^{5,58}.

It should be stressed here again that the main point in this article is that the seemingly subtle difference in cryo-EM map densities of the MOR(DAMGO)-G_i-scFv16 complexes in the absence and presence of BMS-986122 prompted us to focus on these residues, and the importance of the

interaction in the absence of G protein is clearly confirmed by the NMR analyses and GTP turnover assay experiments (Fig. 4). Thanks to the reviewer's suggestion, the current version of the manuscript is more precisely written regarding the interpretation of the difference in the density maps.

Fig. 4 | Importance of R167^{3.50}-Y254^{5.58} hydrogen bonding interaction for function and structural dynamics of MOR

a Intracellular view of structures of MOR bound to DAMGO and BMS-986122 (left) and bound to antagonist β -FNA (right, PDB ID: 4DKL). TM6 helices of MOR were colored green and magenta, and the C ϵ atoms of M283 were represented by spheres.

b ¹H-¹³C HMQC spectra of the parental MOR/ Δ 6M (left) or the Y254^{5.58}F variant (right)

bound to DAMGO in the presence and absence of BMS-986122.

c GTP turnover of the G_i protein stimulated by MOR/Δ6M or the Y254^{5,58}F variant in the DAMGO-bound state in the presence and absence of BMS-986122. All values were normalized by the GTP turnover rate of the G_i protein in the presence of MOR/Δ6M in the full agonist DAMGO-bound state. Data are presented as mean ± standard error of the mean (s.e.m.) (n=3 independent replicates). Statistical significance was determined by a one-tailed Student's t-test. *: p < 0.05.

#2-minor5. The manuscript contains numerous speculative statements, such as the H-bond interaction between R167 and Y254 stabilizing the outward-shifted TM6 in the absence of G protein, without adequate experimental evidence.

We would like to thank the reviewer for their comment regarding the speculative statements in the original manuscript. The outward shift of TM6 in the absence of G proteins has been experimentally demonstrated through NMR analyses, specifically solvent PRE analyses, as reported in our preceding article (Kaneko *et al.*, *Proc Natl Acad Sci USA* (2022) **119**:e2121918119). Thus, the observation of the NMR signal of M283 from the fully-activated conformation (labeled as M283^{FA} in Fig. 2 in Kaneko *et al.*, *Proc Natl Acad Sci USA* (2022) **119**:e2121918119, shown below for your reference) directly shows that a portion of the MOR population in the absence of G protein is in the fully-activated conformation (the conformation with outward-shifted TM6).

[REDACTED]

Fig. 2 in Kaneko *et al.*, *Proc Natl Acad Sci USA* (2022) **119**:e2121918119 | The solvent PRE analyses of MOR in the full agonist DAMGO-bound state. [REDACTED]

in the intensity of the signal relative to the other signals from M283 (M283^I and M283^{PA}). Therefore, the stabilization of the outward-shifted TM6 by the hydroxy group of Y254^{5.58} has been experimentally demonstrated through mutational analyses, where the disappearance of the M283^{FA} signal due to the substitution of Y254^{5.58} with a phenylalanine residue was observed (Fig. 4b). The role of the hydroxy group of Y254^{5.58} in the hydrogen bonding with R167^{3.50} in the absence of G proteins was indeed speculated based on our cryo-EM structure (Fig. 2), as the reviewer pointed out. To complement the Y254^{5.58}F mutagenesis experiments, we performed additional experiments in which R167 was substituted with a leucine residue (Supplementary Fig. 8). As a result, both of the NMR signals from the fully-activated and partially-activated states disappeared for the MOR/Δ6M/R167^{3.50}L variant in the full-agonist DAMGO-bound state, and only the signal from the inactivated state was observed both in the absence and presence of BMS-986122. This indicates that these states are destabilized by the R167^{3.50}L mutation (Supplementary Fig. 8a). The results of the GTP turnover assay were also consistent with this, where the R167^{3.50}L variant exhibited no significant activities both in the absence and presence of BMS-986122, compared to the wild-type (Supplementary Fig. 8b). These results strongly support the involvement of R167^{3.50} in the formation of the fully-activated conformation, where the intracellular half of the TM6 helix is outward shifted. Together with the Y254^{5.58}F mutagenesis experiment and the structural connectivity observed in the cryo-EM structure of MOR(DAMGO)-G_i-ScFv16 in the presence of BMS-986122, we concluded that the interaction between R167^{3.50} and Y254^{5.58} is crucial, supported by substantial experimental evidence.

Fig. 2 | Comparison of structures of MOR(DAMGO) in the presence and absence of BMS-986122

a Comparison of cryo-EM structures of the MOR(DAMGO)-G_i-scFv16 complex in the presence (green) and absence (cyan) of BMS-986122.

b Overlay of cryo-EM density maps and structures around R167^{3.50}, Y254^{5.58}, and Y338^{7.53} residues of MOR(DAMGO) in the presence (left) and absence (right) of BMS-986122. The differences in density between R167^{3.50} and Y254^{5.58} are highlighted with red circles. Densities are contoured at 3.00 and 0.012 in ChimeraX for the maps in the presence and absence of BMS-986122, respectively, representing a significant difference in the contact region between R167^{3.50} and Y254^{5.58}.

Fig. 4 | Importance of R167^{3.50}-Y254^{5.58} hydrogen bonding interaction for function and structural dynamics of MOR

a Intracellular view of structures of MOR bound to DAMGO and BMS-986122 (left) and bound to antagonist β -FNA (right, PDB ID: 4DKL). TM6 helices of MOR were colored green and magenta, and the C ϵ atoms of M283 were represented by spheres.

b ¹H-¹³C HMQC spectra of the parental MOR/ Δ 6M (left) or the Y254^{5.58}F variant (right) bound to DAMGO in the presence and absence of BMS-986122.

c GTP turnover of the G_i protein stimulated by MOR/ Δ 6M or the Y254^{5.58}F variant in the DAMGO-bound state in the presence and absence of BMS-986122. All values were normalized by the GTP turnover rate of the G_i protein in the presence of MOR/ Δ 6M in the full agonist DAMGO-bound state. Data are presented as mean \pm standard error of the mean (s.e.m.) (n=3 independent replicates). Statistical significance was determined by a one-tailed Student's t-test. *: p < 0.05.

Supplementary Fig. 8 | Importance of the R167^{3.50}-Y254^{5.58} interaction for structural dynamics and function of MOR investigated by the R167^{3.50}L variant.

a ¹H-¹³C HMQC spectra of the R167^{3.50}L variant bound to DAMGO in the presence and absence of BMS-986122.

b GTP turnover of G_i protein stimulated by MOR/Δ6M or R167^{3.50}L variant in the DAMGO-bound state in the presence and absence of BMS-986122. All values were normalized by the GTP turnover rate of the G_i protein in the presence of MOR/Δ6M in the full agonist DAMGO-bound state.

In response to the reviewer's comment, we have revised the main text as follows. Thanks to the reviewer's insights, we believe that our conclusions are now adequately supported by the experimental evidence.

Results (P.15, line 257 – P.16, line 279):

For the Y254^{5.58}F variant, in which the interaction between R167^{3.50} and Y254^{5.58} would be abolished, the signals from the fully-activated and inactivated conformations decreased and increased, respectively, both in the absence and presence of BMS-986122.

For the R167^{3.50}L variant, in which the interaction between R167^{3.50} and Y254^{5.58} would also be abolished, similar effects were observed; i.e., the signals from both the fully-

activated and partially-activated conformations decreased, whereas the signal from the inactivated conformation increased (Supplementary Fig. 8a). Together, these findings indicate that the disruptions of the interaction between R167^{3.50} and Y254^{5.58} by introducing the Y254^{5.58}F or R167^{3.50}L mutations destabilize the fully-activated conformation, which features the outward-shifted TM6.

Finally, the contribution of R167^{3.50} and Y254^{5.58} to the activity of the MOR was investigated using a GTP turnover assay³⁹⁻⁴¹ (Fig. 4c and Supplementary Fig. 8b). When normalized by the GTP turnover rate of the G_i protein in the presence of MOR/Δ6M(DAMGO), the activities of the MOR/Δ6M/Y254^{5.58}F(DAMGO) variant were $-4 \pm 1\%$ and $48 \pm 7\%$ in the absence and presence of BMS-986122 (Fig. 4c), and that of the MOR/Δ6M/R167^{3.50}L(DAMGO) variant were $-17.3 \pm 0.6\%$ and $-16.9 \pm 1.6\%$, respectively (Supplementary Fig. 8b). The observation that the activity was partly rescued by the addition of BMS-986122 in the Y254^{5.58}F variant indicated that the Y254^{5.58}F mutation does not abolish the structure of MOR nor the interaction with the G_i protein. Therefore, these results demonstrated that the disruption of the interaction between R167^{3.50} and Y254^{5.58} drastically decreases the G_i protein-stimulating activity of MOR, consistent with the decrease of the population of the fully-activated conformation observed in the NMR experiments (Fig. 4b and Supplementary Fig. 8a).

Fig. 4 | Importance of R167^{3.50}-Y254^{5.58} hydrogen bonding interaction for function and structural dynamics of MOR

a Intracellular view of structures of MOR bound to DAMGO and BMS-986122 (left) and bound to antagonist β -FNA (right, PDB ID: 4DKL). TM6 helices of MOR were colored green and magenta, and the C ϵ atoms of M283 were represented by spheres.

b ¹H-¹³C HMQC spectra of the parental MOR/ Δ 6M (left) or the Y254^{5.58}F variant (right) bound to DAMGO in the presence and absence of BMS-986122.

c GTP turnover of the G_i protein stimulated by MOR/ Δ 6M or the Y254^{5.58}F variant in the DAMGO-bound state in the presence and absence of BMS-986122. All values were normalized by the GTP turnover rate of the G_i protein in the presence of MOR/ Δ 6M in the full agonist DAMGO-bound state. Data are presented as mean \pm standard error of the mean (s.e.m.) (n=3 independent replicates). Statistical significance was determined by a one-tailed Student's t-test. *: p < 0.05.

Supplementary Fig. 8 | Importance of the R167^{3.50}-Y254^{5.58} interaction for structural dynamics and function of MOR investigated by the R167^{3.50}L variant.

a ¹H-¹³C HMQC spectra of the R167^{3.50}L variant bound to DAMGO in the presence and absence of BMS-986122.

b GTP turnover of G_i protein stimulated by MOR/ Δ 6M or R167^{3.50}L variant in the DAMGO-bound state in the presence and absence of BMS-986122. All values were normalized by the GTP turnover rate of the G_i protein in the presence of MOR/ Δ 6M in the full agonist DAMGO-bound state.

Reviewer #3:

The manuscript presents an insightful study on the three-dimensional structure of the μ -opioid receptor (MOR) in complex with the Gi protein and an allosteric modulator, BMS-986122, using cryogenic electron microscopy (cryo-EM). The authors indicate that binding of BMS-986122 induces alterations in map densities associated with R1673.50 and Y2545.58, pivotal residues within conserved structural motifs among class A G-protein-coupled receptors (GPCRs). The findings provide valuable insights into the mechanisms underlying the variations in signaling activity by allosteric modulators. This manuscript is generally well written, the figures are of relatively good quality, and the conclusions are supported by the data. However, there are a few points that should be addressed prior to its acceptance for publication, including rewriting the manuscript and performing some additional experiments, as follows:

We would like to thank the reviewer for their insightful comments and suggestions. We have carefully revised the manuscript in response to their feedback. Below are our responses to each of the comments and suggestions provided by the reviewer.

Reviewer#3-1. The NMR analyses complement the structural findings and add depth to the study. The enhancement of the interaction between R1673.50 and Y2545.58 by BMS-986122, thus stabilizing the fully-activated conformation is a crucial observation. However, NMR experiments were conducted in the absence of G proteins, which differs from the physiological conditions. Therefore, I suggest supplementing the conclusions with additional molecular and cellular-level functional experiments, such as cAMP accumulation assays, NanoBiT experiments, or any other

pertinent methodology of the authors' choice. This would serve to strengthen the aforementioned conclusions.

We appreciate the reviewer's suggestion to conduct additional molecular and cellular level functional experiments to strengthen our conclusions. Accordingly, we have performed a cAMP inhibition assay using mammalian cells, as described in a previous study on the delta opioid receptor (Fenalti *et al.*, *Nature* (2014) 506:191) (Supplementary Fig. 9, shown below). In this assay, split luciferase fused to a cAMP binding domain, known as GloSensor (Promega) is co-expressed with either wild-type MOR or its Y254^{5,58}F variant. The intracellular cAMP concentrations are then monitored by measuring luciferase activity. The activity of MOR is determined by the decrease in luminescence from the control experiments. Although non-specific changes were observed upon the addition of BMS-986122 alone, the Y254^{5,58}F variant of MOR exhibited reduced G_i-stimulating activity compared to the wild-type, both in the absence and presence of BMS-986122. These results strongly support the conclusion that the enhancement of the interaction between R167^{3,50} and Y254^{5,58} by BMS-986122 stabilizes the fully-activated conformation.

Supplementary Fig. 9 | cAMP inhibition assay for wild-type and Y254^{5.58}F variant of MOR/Δ6M

The degrees of G_i activation were quantified as the decrease in the magnitude of luminescence by the cAMP-dependent split luciferase, GloSensor (n=6). Data are presented as mean ± standard error of the mean (s.e.m.). Statistical significance was determined by a two-tailed unpaired Student's t-test. **: p < 0.0005, ns: not significant (p > 0.05).

We have added these data as Supplementary Fig. 9 and referenced it in the main text, as below. Thanks to the reviewer's suggestion, our conclusion from NMR analyses is robustly supported by the cellular experiments under physiological conditions.

Results (P.16, lines 277 – 285):

Therefore, these results demonstrated that the disruption of the interaction between R167^{3.50} and Y254^{5.58} drastically decreases the G_i protein-stimulating activity of MOR, consistent with the decrease of the population of the fully-activated conformation observed in the NMR experiments (Fig. 4b and Supplementary Fig. 8a). This is further supported by the cellular cAMP inhibition assay, where substituting Y254^{5.58} with phenylalanine abolished the activity of MOR (Supplementary Fig. 9). Together, these results indicate that the interaction of R167^{3.50} and Y254^{5.58} is involved in the regulation of the function of MOR via the dynamic conformational equilibrium, and the binding of BMS-986122 enhances the function by shifting that conformational equilibrium.

Methods (P.32, line 577 – P.34, line 597):

cAMP inhibition assay

To measure G_i-mediated cAMP inhibition, 293T cells were co-transfected with wild-type MOR/Δ6M (with the N-terminal sequence that was cleaved by TEV protease in the NMR experiments) or the Y254^{5.58}F variant, along with a luciferase-based cAMP biosensor (GloSensor, Promega)⁵⁴. 1.8 μg of pGloSensor-22F cAMP plasmid (Promega) and 0.2 μg of the pCI-neo vector harboring wild-type MOR/Δ6M, the Y254^{5.58}F variant of MOR/Δ6M, or the empty backbone pCI-neo vector, were mixed and transfected into 5×10⁵ 293T cells cultured in a 6-well plate for 24 hours in DMEM supplemented with 1.0% dialyzed FBS, using X-tremeGENE 360 (Roche). After 24 hours, cells were harvested and resuspended into a 96-well white flat-bottom tissue-culture-treated microplate (Corning) at 2.0×10⁴ cells/well, in CO₂-independent medium supplemented with 1% (v/v) dialyzed FBS and 2% (v/v) GloSensor cAMP Reagent (Promega). After

incubating for 2 hours at room temperature, DAMGO and/or BMS-986122 were added to final concentrations of 1 μM , and the plates were further incubated for 15 minutes at room temperature. Then, isoproterenol (Sigma) was added to a final concentration of 1 μM to activate G_s protein via endogenous β_2 -adrenergic receptors, and luminescence intensity was quantified 15 minutes later with an EnVision 2104 multilabel plate reader (PerkinElmer) for 10 seconds. The degree of G_i activation was quantified by subtracting the luminescence values from control wells, where buffer was added instead of DAMGO or BMS-986122. The experiments were repeated six times, and mean values and standard errors are reported.

References

54. Fenalti, G. *et al.* Molecular control of δ -opioid receptor signalling. *Nature* **506**, 191–6 (2014).

Supplementary Fig. 9 | cAMP inhibition assay for wild-type and Y254^{5.58}F variant of MOR/Δ6M

The degrees of G_i activation were quantified as the decrease in the magnitude of luminescence by the cAMP-dependent split luciferase, GloSensor (n=6). Data are presented as mean ± standard error of the mean (s.e.m.). Statistical significance was determined by a two-tailed unpaired Student's t-test. **: p < 0.0005, ns: not significant (p > 0.05).

Reviewer#3-2a. Lines 125 to 127: Given that the structure has already been solved before, why did the authors determine it again?

We solved the structure of the MOR(DAMGO)-G_i-scFv16 because it was imperative to compare the cryo-EM density maps obtained in the absence and presence of BMS-986122 under otherwise identical conditions, for unambiguously assigning the density from BMS-986122, as well as identifying possible structural differences in MOR. The previously reported MOR structure in the DAMGO-bound state and in complex with the G_i protein was of mouse MOR, which is 98% identical to the human one used here, with differences at I68 (V in mouse MOR), T139 (N), I189 (V), V308(I), and W158 (introduced as a thermostabilizing mutation, F in mouse, see below in Reviewer#3-4 for detail). Furthermore, during the initial screening, it was demonstrated that fusing the BRIL protein at the N-terminus, recently reported for MOR but not in the DAMGO bound state (Wang *et al.*, *Angew. Chem. Int. Ed.* (2022) 61: e202200269, Qu *et al.*, *Nat Chem Biol* (2022) 19:423), significantly improves the solubility and mono-dispersity of the sample.

Therefore, we obtained a set of two cryo-EM data where the BRIL-fused MOR(DAMGO)-Gi-ScFv16 complexes were purified identically, with the only difference being whether BMS-986122 is added or not. Since the fused BRIL at the N-terminus is mobile and averaged out in the single particle analyses, the obtained density maps and PDB models are of MOR(DAMGO) in complex with the G_i protein and scFv16. However, there are differences from the previously reported data of MOR(DAMGO) in complex with the G_i protein and scFv16; i.e., our construct is the human MOR with the F158^{3.41}W mutation and has a BRIL tag fused at the N-terminus.

Reviewer#3-2b. Are the two structures identical or any discernible differences?

Although different in the primary sequence (Reviewer#3-2a), the structures of MOR(DAMGO)-Gi-ScFv16 reported previously (Koehl *et al.*, *Nature* (2018) 558:547, PDB ID: 6DDF) and obtained

here are virtually identical with the C α atom r.m.s.d of 0.71 Å (Supplementary Fig. 3).

Supplementary Fig. 3 | Comparison of the structures of MOR(DAMGO)-G_i-scFv16

a Overlay of the two cryo-EM structures of MOR(DAMGO)-G_i-scFv16. The structure of the human MOR construct obtained in this study is colored, whereas the previous structure of the mouse MOR construct (PDB ID: 6DDE) is shown in gray. The r.m.s.d of C α atoms of MOR molecules is 0.71 Å.

b A closeup view of Y254^{5,58} and R167^{3,50}. Side chain atoms of the two structures are shown in stick models.

Reviewer#3-2c. Please include additional descriptive elements to illustrate.

We thank the reviewer for the suggestion to include additional descriptive elements to illustrate the difference between the previously reported and current structures. Accordingly, we added Supplementary Fig. 3, and the following sentences to the main text. Thanks to the reviewer's suggestion, the differences between these two structures are now clearly described.

Results (P.8, lines 136 – 142):

For a direct comparison with the structure in the presence of BMS-986122, we also obtained a cryo-EM structure of the human MOR(DAMGO)-Gi-scFv16 complex in the absence of BMS-986122 at 3.0 Å resolution (Supplementary Fig. 2 and Supplementary Table 1), which was virtually identical to the previously reported structure of mouse MOR without the N-terminal BRIL tag³³ with the root mean square deviation (r.m.s.d.) of the C α atoms of 0.71 Å (Supplementary Fig. 3).

Supplementary Fig. 3 | Comparison of the structures of MOR(DAMGO)-G_i-scFv16

a Overlay of the two cryo-EM structures of MOR(DAMGO)-G_i-scFv16. The structure of the human MOR construct obtained in this study is colored, whereas the previous structure of the mouse MOR construct (PDB ID: 6DDE) is shown in gray. The r.m.s.d of C α atoms of MOR molecules is 0.71 Å.

b A closeup view of Y254^{5.58} and R167^{3.50}. Side chain atoms of the two structures are shown in stick models.

References

33. Koehl, A. *et al.* Structure of the μ -opioid receptor-Gi protein complex. *Nature* **558**,

547–552 (2018).

Reviewer#3-3. Lines 142 to 145: The structures of BMS-986124 and BMS-986122 exhibit remarkable similarity, yet only BMS-986122 acts as a positive allosteric modulator for MOR. Can authors speculate on the potential reasons for this phenomenon based on the current structural information available?

Yes, we can speculate on the potential reasons for the differences in activity between these two closely related compounds based on our structure. The difference between BMS-986122 and BMS-986124, the non-allosteric modulator derivative of BMS-986122, lies in the arene substitution pattern of the methoxybenzyl bromide moiety (Fig. R5). In BMS-986122, the methoxy and bromide groups are positioned at the para and meta positions, respectively, whereas in BMS-986124, these groups are located at the ortho and para positions, respectively. In the structure of MOR(DAMGO)-G_i-scFv16, the bromide group of BMS-986122 is directed toward TM3 of MOR, potentially stacking with W158^{3,41}. Since the bromide group at the para position in BMS-986124 is not compatible with this interaction, we speculate that a bromide group at the meta position, which is absent in BMS-986124, is crucial for the allosteric modulation of MOR by BMS-986122.

BMS-986122
(allosteric modulator)

BMS-986124
(binds to MOR with
no allosteric modulation)

Fig. R5 | Binding site of BMS-986122 with MOR

In response to the reviewer's comment, we revised Supplementary Fig. 4 (Supplementary Fig. 3 in the original version), and referred to this figure in the main text, as follows.

Results (P.10 lines 168 - 169):

We then ~~modeled~~ docked BMS-986122 in the cryo-EM structure so that the methoxybenzyl bromide moiety is adjacent to M245^{5.49} (Supplementary Fig. 4d).

Supplementary Fig. 4 | Binding site and interaction of BMS-986122

a Two possible orientations of BMS-986122 to MOR. The C ϵ atom of M245^{5.49} is represented by a sphere. The methoxy (blue) and bromo (red) groups are indicated by squares.

b The chemical structures of BMS-986122 (left) and BMS-986124 (right). The methoxy (blue) and bromo (red) groups are indicated by squares.

c ¹H-¹³C HMQC signals of M245^A of MOR/ Δ 6M in DAMGO-bound (left), DAMGO

and BMS-986122 bound (middle), and DAMGO and BMS-986124 bound (right) states. The M245^A signal was not observed for DAMGO and BMS-986122 bound MOR/ Δ 6M.

d A closeup view of BMS-986122 in the cryo-EM structure of MOR(DAMGO)-G_i-scFv16 docked by using the information obtained from the NMR analyses (**b** and **c**).

Reviewer#3-4. In the provided "wwPDB validation report" files, the second section "Entry composition" displays that the amino acid at position 158 in the R chain is modeled as Trp, while it is actually Phe. I did not find any mention of this mutation in the manuscript. Please provide clarification or include a statement addressing this discrepancy in the manuscript.

We would like to express our appreciation to the reviewer for their comment. The F158W mutation has been described at the beginning of the Results section in the original manuscript, as follows:

Results (P. 7, lines 9 - 13 in the original manuscript):

In this study, we used an N-terminally BRIL-tagged human-MOR construct (19, 20) with a F158W mutation, the heterotrimeric G_i protein, and a single chain antibody scFv16 (21) for the cryo-EM study, and a human-MOR construct with F158W and six methionine substitutions, referred to as MOR/ Δ 6M, for the NMR study (4, 15) (see Methods for detail).

The F158^{3,41}W mutation is a so-called W^{3,41} mutation in class A GPCRs, which is known to increase thermostability as well as total expression levels, including in MOR (Heydenreich *et al.*, *Front Pharmacol.* (2015) 6:82, Zhuang *et al.*, *Cell* (2022) 185:4361). We have utilized this mutation in our preceding articles and confirmed that it does not abolish MOR activity or sensitivity to BMS-986122

(Okude *et al.*, *Angew. Chem. Int. Ed.* (2015) 54:15771, and Kaneko *et al.*, *Proc. Natl. Acad. Sci. USA* (2022) 119:e2121918119). In accordance with the reviewer's suggestion, we have added the following description to the manuscript. Thanks to the reviewer's suggestion, the use of the F158^{3,41}W mutation is now clearly explained.

Results (P.8, lines 128 – 132)

In this study, we utilized an N-terminally BRIL-tagged^{29,30} human MOR construct with an F158^{3,41}W mutation^{31,32}, the heterotrimeric G_i protein, and a single-chain antibody scFv16³³ for the cryo-EM study, and a human MOR construct with F158^{3,41}W and six methionine substitutions, referred to as MOR/Δ6M, for the NMR study^{4,19} (see Methods for details).

Methods (P.22, lines 382 – 391):

For cryo-EM analyses, a modified human MOR construct with a removable N-terminal Flag-tag (DYKDDDDA) and C-terminal octahistidine tag (HHHHHHHH) was used in this study, as previously described^{29,30}, with minor modifications. N-terminal residues (M1–G63) of MOR were replaced with the thermostabilized apocytochrome b562RIL from *Escherichia coli* (M7W, H102I, and R106L) (BRIL) protein and a linker sequence (GSPGARSAS). C-terminal residues (Q363–P400) of MOR were truncated. The N-terminal Flag-tag and C-terminal histidine tag were removable with rhinovirus 3C protease. **The thermostabilizing mutation F158^{3,41}W, which does not interfere with the activity, was introduced to increase thermostability as well as the total expression level^{31,32}.**

References

29. Wang, H. *et al.* Structure-Based Evolution of G Protein-Biased μ -Opioid Receptor Agonists. *Angew Chem Int Ed Engl* **61**, e202200269 (2022).
30. Qu, Q. *et al.* Insights into distinct signaling profiles of the μ OR activated by diverse agonists. *Nat Chem Biol* **19**, 423–430 (2023).
31. Heydenreich, F. M., Vuckovic, Z., Matkovic, M. & Veprintsev, D. B. Stabilization of G protein-coupled receptors by point mutations. *Front Pharmacol* **6**, 82 (2015).
32. Zhuang, Y. *et al.* Molecular recognition of morphine and fentanyl by the human μ -opioid receptor. *Cell* **185**, 4361-4375.e19 (2022).

Reviewer#3-5. Some abbreviations are not explained when they first appear in the text, for example, line 98, “cryo-EM” and line 145, “HMQC” are not explained.

We would like to thank the reviewer for pointing out that some abbreviations were not explained when they first appeared in the manuscript. We have reviewed the entire text and added explanations for the abbreviations as follows. Thanks to the reviewer’s comment, the manuscript has become more comprehensive for the broad readership of *Nature Communications*.

Results (P.6, line 107 – P.7, line 110):

Here, we determine the three-dimensional structure of MOR in complex with the heterotrimeric G_i protein, in a full agonist [d-Ala2,N-MePhe4,Gly-ol5]enkephalin (DAMGO)- and the allosteric modulator BMS-986122-bound state, using cryo-electron microscopy (cryo-EM).

Results (P.9, lines 160 – 162):

When the ^1H - ^{13}C heteronuclear multiple quantum coherence (HMQC) NMR spectra of

MOR(DAMGO) in the presence of BMS-986122 and BMS-986124 were compared, the NMR signal from M245^{5.49} was largely perturbed;...

Reviewer#3-6. The numbering of amino acids lacks consistency. Please standardize it according to the Ballesteros-Weinstein numbering system. For instance, line 181, R167 should be revised to R167^{3.50}.

We apologize for the inconsistency in amino acid numbering. In accordance with the reviewer's suggestion, we have now described all residue numbering using the Ballesteros-Weinstein numbering system, such as R167^{3.50}. Furthermore, we have standardized this numbering scheme throughout the residues in Figures and Supplementary Figures. Please note that we have kept the labeling for NMR signals as is to avoid confusion. Thanks to the reviewer's suggestion, the residue numbers in the manuscript are now standardized and consistent.

Fig. 2 | Comparison of structures of MOR(DAMGO) in the presence and absence of BMS-986122

a Comparison of cryo-EM structures of the MOR(DAMGO)-G_i-scFv16 complex in the presence (green) and absence (cyan) of BMS-986122.

b Overlay of cryo-EM density maps and structures around R167^{3.50}, Y254^{5.58}, and Y338^{7.53} residues of MOR(DAMGO) in the presence (left) and absence (right) of BMS-986122. The differences in density between R167^{3.50} and Y254^{5.58} are highlighted with red circles. Densities are contoured at 3.00 and 0.012 in ChimeraX for the maps in the presence and absence of BMS-986122, respectively, representing a significant difference in the contact region between R167^{3.50} and Y254^{5.58}.

Fig. 3 | M257^{5.61} methyl NMR signal as a monitor for R167^{3.50}-Y254^{5.58} interaction

a A closeup view of the cryo-EM structure of MOR(DAMGO) in complex with BMS-986122. The C ϵ atom of M257^{5.61} is represented by a sphere.

b ^1H - ^{13}C HMQC spectra of MOR(DAMGO) in the presence and absence of BMS-986122 at the ^1H frequencies of 800 MHz and 1 GHz. Cross sections of the signal of M257^{5.61} in the ^{13}C direction are shown in black. Colored lines show the peaks obtained from spectral deconvolution of the cross sections. Linewidths of the M257 signal in the ^{13}C direction are shown with blue highlights.

c Differences in linewidths of the M257 signal between 800 MHz and 1 GHz. Error bars represent the estimated deviations from the line shape fitting.

d Overlays of ^1H - ^{13}C HMQC spectra and cross section in the ^{13}C direction of the M257 signal of parental MOR/Δ6M (black) and the Y254^{5.58}F variant (red).

Fig. 4 | Importance of R167^{3.50}-Y254^{5.58} hydrogen bonding interaction for function and structural dynamics of MOR

a Intracellular view of structures of MOR bound to DAMGO and BMS-986122 (left) and bound to antagonist β -FNA (right, PDB ID: 4DKL). TM6 helices of MOR were colored green and magenta, and the C ϵ atoms of M283 were represented by spheres.

b ¹H-¹³C HMQC spectra of the parental MOR/ Δ 6M (left) or the Y254^{5.58}F variant (right) bound to DAMGO in the presence and absence of BMS-986122.

c GTP turnover of the G_i protein stimulated by MOR/ Δ 6M or the Y254^{5.58}F variant in

the DAMGO-bound state in the presence and absence of BMS-986122. All values were normalized by the GTP turnover rate of the G_i protein in the presence of MOR/ $\Delta 6M$ in the full agonist DAMGO-bound state. Data are presented as mean \pm standard error of the mean (s.e.m.) (n=3 independent replicates). Statistical significance was determined by a one-tailed Student's t-test. *: $p < 0.05$.

Fig. 5 | The role of R167^{3.50}-Y254^{5.58} hydrogen bonding interaction in the function-related equilibrium of MOR

A schematic model illustrating the effects of the hydrogen bonding interaction between R167^{3.50} and Y254^{5.58} on the conformational equilibrium of MOR is shown.

Reviewer#3-7. In Figure 1, panel A, please provide threshold level for the density under the current view, and please check all figures in the manuscript.

We appreciate the reviewer's comment on providing threshold levels for the density under the

current view. In accordance with the comment, we have added the threshold level used to the legend of Fig. 1a, and all other figures where the density of cryo-EM maps is shown. Thanks to the reviewer's comment, the figures can now be readily reproducible from the provided density map.

Fig. 1 | Overall structure of MOR in complex with a full agonist and an allosteric modulator

a A cryo-EM density map of the MOR(DAMGO)-G_i-scFv16 complex in the presence of BMS-986122. Insets show expanded views of the DAMGO and BMS-986122 models and their corresponding density maps. **Densities are contoured at a threshold level of 2.4 in ChimeraX.**

b The overall structure of the MOR(DAMGO)-G_i-scFv16 complex in the presence of BMS-986122.

Fig. 2 | Comparison of structures of MOR(DAMGO) in the presence and absence of BMS-986122

a Comparison of cryo-EM structures of the MOR(DAMGO)-G_i-scFv16 complex in the presence (green) and absence (cyan) of BMS-986122.

b Overlay of cryo-EM density maps and structures around R167^{3.50}, Y254^{5.58}, and Y338^{7.53} residues of MOR(DAMGO) in the presence (left) and absence (right) of BMS-986122. The differences in density between R167^{3.50} and Y254^{5.58} are highlighted with red circles. Densities are contoured at 3.00 and 0.012 in ChimeraX for the maps in the presence and absence of BMS-986122, respectively, representing a significant difference in the contact region between R167^{3.50} and Y254^{5.58}.

Reviewer#3-8. In Figure 3, Please maintain uniformity in the formatting of the y-axis labels for both the B panel and D panel.

We appreciate the reviewer's comment to maintain uniformity in the formatting of the y-axis labels for both Fig. 3b and 3d. We have done so by adding the label ' ^{13}C (ppm)' to all spectra in Fig. 3b and 3d.

Fig. 3 | M257^{5.61} methyl NMR signal as a monitor for R167^{3.50}-Y254^{5.58} interaction

a A closeup view of the cryo-EM structure of MOR(DAMGO) in complex with BMS-986122. The C ϵ atom of M257^{5.61} is represented by a sphere.

b ^1H - ^{13}C HMQC spectra of MOR(DAMGO) in the presence and absence of BMS-986122 at the ^1H frequencies of 800 MHz and 1 GHz. Cross sections of the signal of M257^{5.61} in the ^{13}C direction are shown in black. Colored lines show the peaks obtained from spectral deconvolution of the cross sections. Linewidths of the M257 signal in the

¹³C direction are shown with blue highlights.

c Differences in linewidths of the M257 signal between 800 MHz and 1 GHz. Error bars represent the estimated deviations from the line shape fitting.

d Overlays of ¹H-¹³C HMQC spectra and cross section in the ¹³C direction of the M257 signal of parental MOR/Δ6M (black) and the Y254^{5,58}F variant (red).

#3-9. Line 341, “MWCO 10K” should be revised as “MWCO 10KDa”

We appreciate the reviewer’s comment and have modified the description as follows. We used 10 kDa instead of KDa, which is consistent with the other descriptions in the manuscript.

Methods (P.23, lines 413 – 415):

The cells were spun down by centrifugation at 2,000 rpm for 15 min, and the supernatant was concentrated about three-fold with a pressure-based sample concentration device (MWCO 10 **kDa**).

Reviewer#3-10. Line 433, “PHHHHHHHH” should be revised as “HHHHHHHHH”

We appreciate the reviewer’s comment and have modified the description as follows:

Methods (P.29, lines 507 – 509):

For NMR analyses, a modified human MOR construct with a removable N-terminal Flag-tag (DYKDDDDA) and a nonremovable C-terminal 8x histidine tag (**HHHHHHHH**), MOR/Δ6M, was used as previously described¹⁹.

Reviewer#3-11. Please ensure consistency in the citation format, for instance, line 541, DOI is not required.

We appreciate the reviewer's comment. In accordance with the suggestion, we ensured consistency in the citation format.

REVIEWERS' COMMENTS

Reviewer #1 (Remarks to the Author):

I've reviewed the paper and the detailed rebuttal plus additional experiments, qualifiers, and experiments. I think the authors did a good job addressing most of my concerns and distinguishing this work from their earlier PNAS article.

Reviewer #2 (Remarks to the Author):

The authors have satisfactorily addressed most of my concerns.

Reviewer #3 (Remarks to the Author):

The revised manuscript is much improved and my concerns have been mostly addressed. I would therefore recommend its publication after addressing the minor issues listed below:

1. Line 200, "is" should be corrected to "was". Please check the tenses throughout the manuscript.
2. Line 298, "293T" should be corrected to "HEK293T".
3. Since the full name of "PDB" has been previously mentioned in the text, "Protein Data Bank (PDB)" in line 492 should be revised to "PDB".
4. There is no journal name in reference 28. Please check the reference format carefully.

Point-to-point responses to the reviewer comments

We would like to express our sincerest gratitude to all the referees for their prompt reviewing of our manuscript. Below, we provide our point-by-point responses to the reviewers' comments.

Reviewer #1 (Remarks to the Author):

I've reviewed the paper and the detailed rebuttal plus additional experiments, qualifiers, and experiments. I think the authors did a good job addressing most of my concerns and distinguishing this work from their earlier PNAS article.

We would like to express our gratitude to the reviewer for their positive evaluation of our revised manuscript. Thanks to their insightful and constructive feedback, our manuscript has significantly improved.

Reviewer #2 (Remarks to the Author):

The authors have satisfactorily addressed most of my concerns.

We appreciate the reviewer for reviewing our revised manuscript. Thanks to their insightful and constructive feedback, our manuscript has significantly improved.

Reviewer #3 (Remarks to the Author):

The revised manuscript is much improved and my concerns have been mostly addressed. I would

therefore recommend its publication after addressing the minor issues listed below:

We would like to express our gratitude to the reviewer for reviewing our revised manuscript, and for highlighting the issues remaining therein. We have addressed these points in the final version of the manuscript. Their insightful and constructive feedback has been invaluable in enhancing the quality of our work.

1. Line 200, “is” should be corrected to “was”. Please check the tenses throughout the manuscript.

We would like to thank the reviewer for highlighting the grammatical error. The word “is” in line 200 has been corrected to “was”. We have also checked the tenses throughout the manuscript.

2. Line 298, “293T” should be corrected to “HEK293T”.

Thank you for pointing this out. The cell line name 293T has been corrected to HEK293T throughout the manuscript to ensure consistency and accuracy.

3. Since the full name of “PDB” has been previously mentioned in the text, “Protein Data Bank (PDB)” in line 492 should be revised to “PDB”.

Thank you for pointing this out. The phrase “Protein Data Bank (PDB)” in line 492 has been revised to “PDB” to avoid repetition.

4. There is no journal name in reference 28. Please check the reference format carefully.

Thank you for pointing out the omission in reference 28. We have now included the missing journal name and have carefully reviewed the format for accuracy and completeness.